# Global gradients in intertidal species richness and functional groups

**Jakob Thyrring[1,2,3,4,5]\*, Lloyd S Peck[1]**

[1]British Antarctic Survey, Cambridge, United Kingdom; [2]Department of Zoology, University of British Columbia, Vancouver, Canada; [3]Arctic Research Centre, Department of Bioscience, Aarhus University, Silkeborg, Denmark; [4]Homerton College, University of Cambridge, Cambridge, United Kingdom; [5]Marine Ecology, Department of Bioscience, Aarhus University, Silkeborg, Denmark

**Abstract** Whether global latitudinal diversity gradients exist in rocky intertidal α-diversity and across functional groups remains unknown. Using literature data from 433 intertidal sites, we investigated α-diversity patterns across 155° of latitude, and whether local-scale or global-scale structuring processes control α-diversity. We, furthermore, investigated how the relative composition of functional groups changes with latitude. α-Diversity differed among hemispheres with a mid-latitudinal peak in the north, and a non-significant unimodal pattern in the south, but there was no support for a tropical-to-polar decrease in α-diversity. Although global-scale drivers had no discernible effect, the local-scale drivers significantly affected α-diversity, and our results reveal that latitudinal diversity gradients are outweighed by local processes. In contrast to α-diversity patterns, species richness of three functional groups (predators, grazers, and suspension feeders) declined with latitude, coinciding with an inverse gradient in algae. Polar and tropical intertidal data were sparse, and more sampling is required to improve knowledge of marine biodiversity.

**\*For correspondence:**
jakyrr57@bas.ac.uk

**Competing interests:** The authors declare that no competing interests exist.

## Introduction

The latitudinal diversity gradient in species richness across ecosystems and various functional groups has been a major research topic that has intrigued scientists since at least *Darwin, 1859* and *Wallace, 1878*. Over time, many hypotheses have been proposed to explain this seemingly general ecological pattern in marine and terrestrial ecosystems. Ecological and physical hypotheses dominate the discussions and include drivers such as habitat area, stability, speciation rates, energy availability, and temperature (*Willig et al., 2003*; *Clarke and Gaston, 2006*; *Edgar et al., 2017*). However, the geographic, functional, and taxonomic generality of latitudinal diversity gradients remain lively debated as unimodal, bimodal, and inverse gradients emerge across clades, habitats, and latitudes (*Rivadeneira et al., 2002*; *Waller, 2008*; *Chaudhary et al., 2016*; *Kinlock et al., 2018*). In the marine realm, latitudinal gradients in some groups have been shown to be closely related to oceanographic covariates, such as water temperature (*Roy et al., 2000*), yet mammal richness peaks at high latitudes (*Grady et al., 2019*), and more species have been reported from polar soft-sediment habitats than at many lower latitudes (*Vause et al., 2019*).

While gradients in species diversity have received most attention, latitudinal changes across different functional groups (evaluated here by food acquisition, see Materials and methods) remain less studied despite their importance for ecosystem functioning. Macroalgal canopies shelter understory species from environmental stress (*Krause-Jensen et al., 2016*; *Sejr et al., 2021*), creating protective microhabitats that increase environmental heterogeneity and biodiversity, thereby maintaining a diversified understory assemblages (*Bulleri et al., 2002*; *Watt and Scrosati, 2013*; *Piazzi et al., 2018*). Suspension feeders are important benthic-pelagic energy couplers (*Gili and Coma, 1998*),

and predation is widely accepted as a central structuring process in the composition and abundance of species (*Vermeij, 1987*; *Stanley, 2008*). Latitudinal studies have shown that suspension feeders dominate benthic systems in fully marine environments at high latitudes (*Gili and Coma, 1998*; *De Broyer et al., 2014*), while diversity of coastal macroalgae peaks at mid-latitudes (*Keith et al., 2014*), and predation pressure decreases with latitude and depth from shallow shelves to deep oceans (*Taylor and Taylor, 1977*; *Harper and Peck, 2016*). Most functional group studies have focused on a narrow set of taxa, and global-scale assemblage-wide investigations encompassing both hemispheres are rare. Thus, studies demonstrating global patterns in various functional groups are needed to understand spatial patterns, biological interactions, and ecosystem resilience to climate change.

Intertidal shores rank as one of the most studied marine habitats and are often seen as harbingers for the effects of climate change and invasive species. Regional-scale intertidal studies have found richness gradients of gastropods along coastlines in the eastern Pacific Ocean (*Rivadeneira et al., 2015*; *Fenberg and Rivadeneira, 2019*). However, no latitudinal diversity gradient of gastropods (*Miloslavich et al., 2013*) or macroalgae (*Konar et al., 2010*) was found on a global scale across oceans, and assemblage-wide studies have found missing (*Blanchette et al., 2008*; *Cruz-Motta et al., 2010*) or inverse (*Griffiths and Waller, 2016*) gradients. Conflicting and missing gradients suggest that richness is determined by regional or local (and not global) scale processes through biological interactions and small scale overlapping environmental gradients. For example, high habitat heterogeneity creates environmental stress mosaics that are more important in shaping biological patterns than latitudinal environmental gradients (*Helmuth et al., 2006*; *Jurgens and Gaylord, 2018*). However, the large-scale intertidal studies necessary to evaluate global-scale patterns and processes are generally missing (but see *Cruz-Motta et al., 2010*; *Miloslavich et al., 2013*).

Global assessments of intertidal biodiversity have been hindered by data scarcity from polar intertidal shores, and a lack of estimates of intertidal geographic areas. In fact, only the extent of intertidal mud flats have been quantified on a large scale (*Murray et al., 2019*). However, in recent decades intertidal diversity data from both polar regions have advanced (*Weslawski et al., 2010*; *Griffiths and Waller, 2016*; *Thyrring et al., 2017*; *Thyrring et al., 2021*; *Peck, 2018*), providing novel opportunities to study biodiversity patterns and variation in specific functional groups on a worldwide scale. In this study, we examined latitudinal diversity gradients in local intertidal species richness (termed α-diversity) and four functional groups from 433 locations throughout the northern and southern hemispheres, from the high Arctic to the Antarctic. Intertidal ecosystems comprises a variety of habitats ranging from tidal flats to mangrove forests, each supporting unique communities and taxa. Therefore, to control for habitat effects, we specifically focus on rocky shores and boulder fields, as they represent the most studied intertidal habitats and are present in all oceans. We aim to examine patterns in intertidal α-diversity and functional groups across 155° of latitude, and test the following hypotheses: (1) α-diversity decreases with increasing latitude, as evident in other ecosystems and (2) biodiversity is predicted by local-scale structuring processes rather than global-scale oceanographic features.

## Results

### Latitudinal α-diversity gradients

We extracted data from 433 intertidal sites between 74.8°S and 80.5°N (*Figure 1*). Only 11 sites were located at latitudes above 70°, south and north combined (*Supplementary file 1*). A generalized additive mixed effect model (GAMM) indicated a non-linear relationship among α-diversity and latitude in the northern hemisphere where α-diversity peaked at mid-latitudes between 30 and 55°N ( GAMM, edf = 2.005, p=0.023). While no latitudinal scale environmental driver was significant (*Table 1*), α-diversity was similar in the tropical and Arctic regions (*Figure 2*). In the southern hemisphere, a generalized linear mixed effect model (GLMM) indicated that α-diversity was not related to any latitudinal scale environmental drivers (*Table 1*), and displayed a linear latitudinal pattern with a non-significant slight decline at the highest latitude ( *Table 1*; *Figure 2*).

Local-scale models revealed significant relationships between α-diversity and the four local-scale variates ice scour, macroalgal canopy cover, salinity, and wave exposure (*Table 2*). We focused

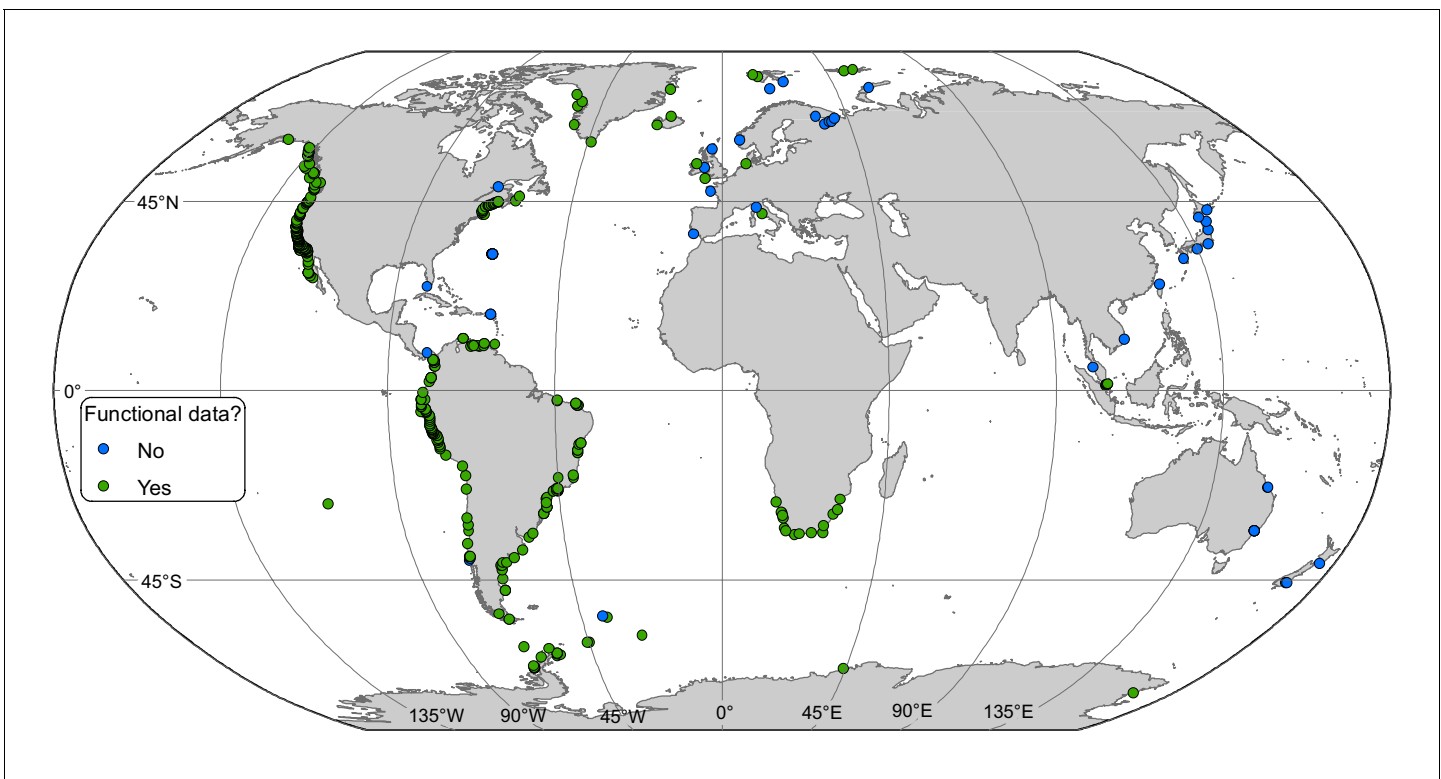

**Figure 1.** Locations of rocky intertidal sampling sites. Some sites are not visible because of close proximity. Functional diversity data were available in green sites.

exclusively on canopy-forming algae because canopies provide living space and protection from predation and extreme temperatures, thereby increasing α-diversity and coverage of understory organisms. A positive relationship was identified between α-diversity and salinity and macroalgal cover (*Figure 3a,c*), with salinity having the strongest effect ($R^2$ = 0.19, 95% CI = 4.82–8.52, *Figure 4*). Wave exposure and ice scour had negative effects on α-diversity (*Figure 3b,d*). The non-linear relationship between wave exposure and α-diversity showed that high levels of wave exposure were

**Table 1.** Results of the mixed effect models for rocky intertidal α-diversity.

The estimate, standard error (SE), z-value, and p-value are presented for each variable. The estimated degree of freedom (ed), and p-value are presented for the smoother in the northern hemisphere*.

|  | Estimates | SE | z-Value | p-Value |
|---|---|---|---|---|
| Northern hemisphere |  |  |  |  |
| (Intercept) | 3.086 | 0.283 | 10.92 | <0.001 |
| Chlorophyll *a* | −0.004 | 0.088 | 0.05 | 0.959 |
| Phosphate | −0.163 | 0.288 | −0.56 | 0.573 |
| Sea temperature | 0.024 | 0.015 | 1.64 | 0.101 |
| Southern hemisphere |  |  |  |  |
| (Intercept) | 3.199 | 0.443 | 7.225 | <0.001 |
| Latitude | 0.006 | 0.007 | 0.899 | 0.369 |
| Chlorophyll *a* | −0.006 | 0.007 | −0.829 | 0.407 |
| Phosphate | −0.240 | 0.234 | −1.029 | 0.303 |
| Sea temperature | −0.010 | 0.016 | −0.653 | 0.514 |

*Approximate significance of cubic spline regression smoother: Estimated df = 2.005, p-value=0.023.

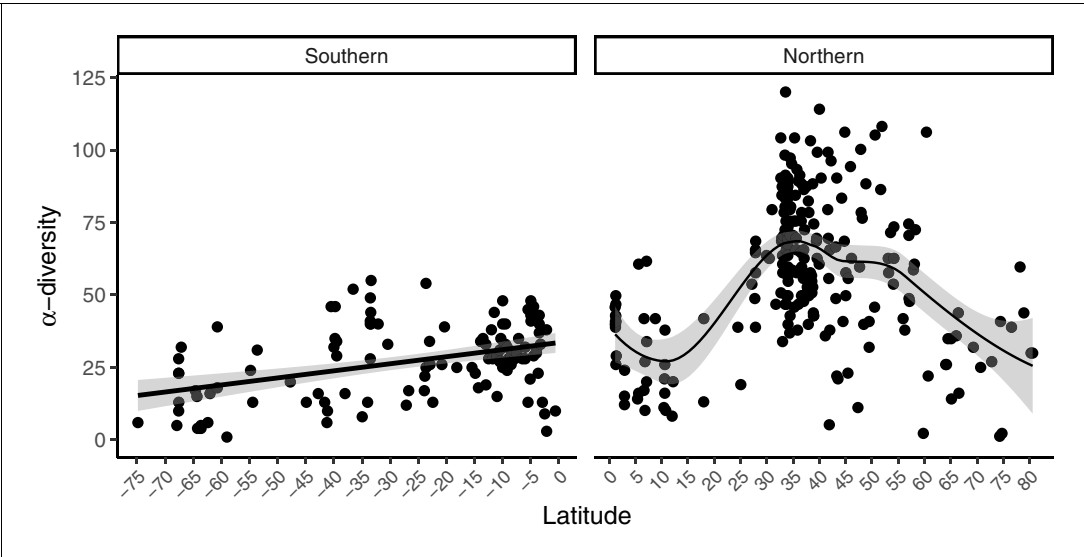

**Figure 2.** Latitudinal patterns in rocky intertidal α-diversity plotted against latitude. Data are split into southern and northern hemispheres. A linear regression line (southern hemisphere) and a best-fit locally weighted scatterplot smoother (northern hemisphere) was added with 95% confidence intervals to aid visual interpretation.

necessary to affect levels of α-diversity (*Figure 3b*). Fewer species inhabited ice-scoured shores compared to ice-free shores (95% CI = −0.57 − −0.23, *Table 2*), although the model only explained around 5% ($R^2$ = 0.05) of the variation due to large differences in the number of species on both ice-scoured and ice-free shores (*Figure 3c*).

**Table 2.** Local-scale model summaries with individual models indicated for each model.
Estimated parameters, standard error (SE), bootstrapped 95% confidence intervals (95% CI), z-values, and p-values are reported for the relationship between α-diversity and environmental covariates.

**Salinity**
**$R^2$ = 0.19 (GLM)**

|  | Estimates | SE | 95% CI | z-Value | p-Value |
|---|---|---|---|---|---|
| (Intercept) | −3.201 | 0.891 | −4.94; −1.47 | −3.6 | <0.0001 |
| Salinity | 6.676 | 0.946 | 4.82; 8.52 | 7.06 | <0.0001 |

**Wave exposure (GAM)**
**$R^2$ = 0.12**

|  | Estimates | SE |  | z-Value | p-Value |
|---|---|---|---|---|---|
| (Intercept)* | 2.435 | 0.029 |  | 82.4 | <0.0001 |

**Ice scour (GLM)**
**$R^2$ = 0.05**

|  | Estimates | SE | 95% CI | z-Value | p-Value |
|---|---|---|---|---|---|
| (Intercept) | 2.353 | 0.055 | 2.24; 2.46 | 42.7 | <0.0001 |
| Ice scour | −0.401 | 0.087 | −0.57; −0.23 | −4.59 | <0.0001 |

**Macroalgal cover (GLM)**
**$R^2$ = 0.14**

|  | Estimates | SE | 95% CI | z-Value | p-Value |
|---|---|---|---|---|---|
| (Intercept) | 6.133 | 0.386 | 5.37; 6.89 | 15.9 | <0.0001 |
| Cover | −2.35 | 0.547 | −3.42; −1.28 | −4.3 | <0.0001 |

*Approximate significance of locally weighted scatterplot wave exposure smoother: Estimated df = 8.40, p-value<0.0001.

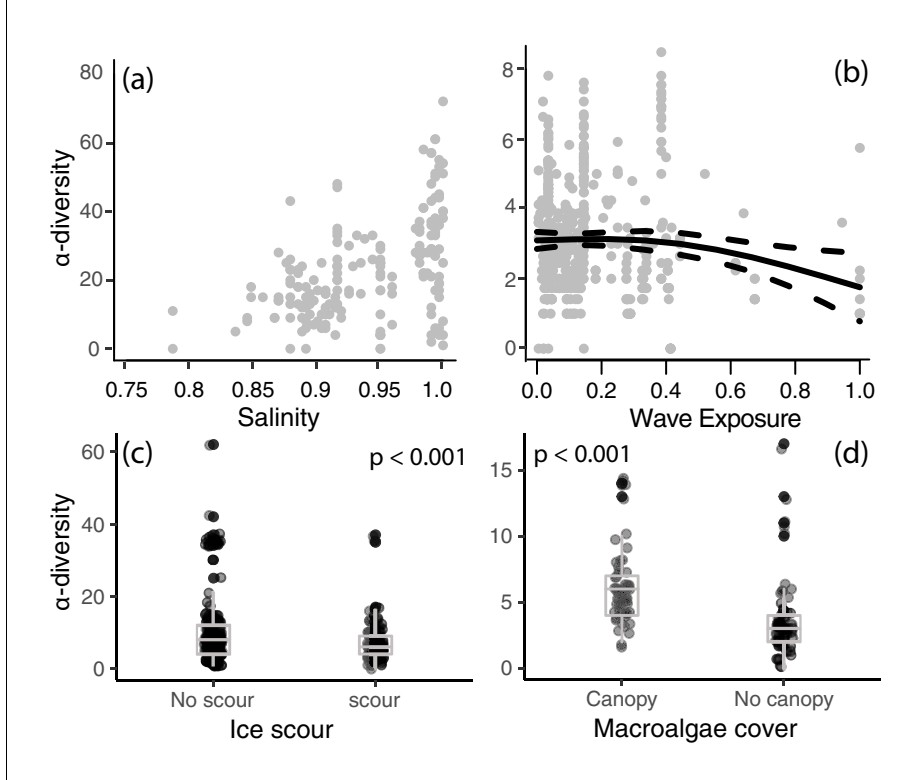

**Figure 3.** Local-scale relationships between (**a**) salinity, (**b**) wave exposure, (**c**) ice scour, and (**d**) macroalgal canopy cover on α-diversity. A best-fit locally weighted scatterplot smoother (LOESS) and 95% confidence intervals (panel **b**) and boxplots (panels **c,d**) has been added to aid visual interpretation.

## Latitudinal gradients in functional groups

Four studied functional groups (algae, grazers, predators, and suspension feeders) displayed different latitudinal patterns (*Figure 5*). Predators, grazers, and suspension feeders declined with latitude in both hemispheres, coinciding with an increase in algal richness (*Figure 5*). In general, the relative distribution of the functional groups changed faster across latitudes in the southern hemisphere where rapid changes were observed around 20°S, while northern hemisphere communities were more consistent from 25 to 60°N (*Figure 5*). We were unable to describe functional groups at latitudes 15–25°N as no data were available (indicated as white bars in *Figure 5*).

In the southern hemisphere, the proportion of predators and grazers declined beyond around 20°S, together forming 6.9% (±6.6 s.d.) of species accounted for between 20 and 55°S, but constituting 18.3% (±15.1 s.d.) of species in communities at latitudes less than 20°S (*Figure 5*). Suspension feeders were common in low- to mid-latitudes, constituting 34.6% (±9.6 s.d.) of the species present between 20 and 45°S

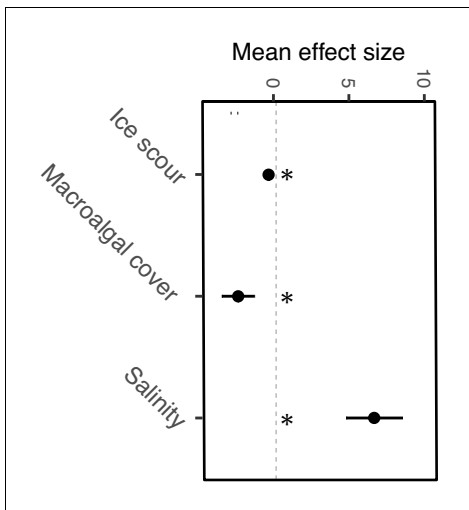

**Figure 4.** Local-scale mean effect sizes and direction of ice scour, macroalgal cover, and salinity on α-diversity estimated from individual models. Significance of regression parameters is identified as bootstrapped 95% confidence intervals (error bars) not crossing zero (* indicates significance).

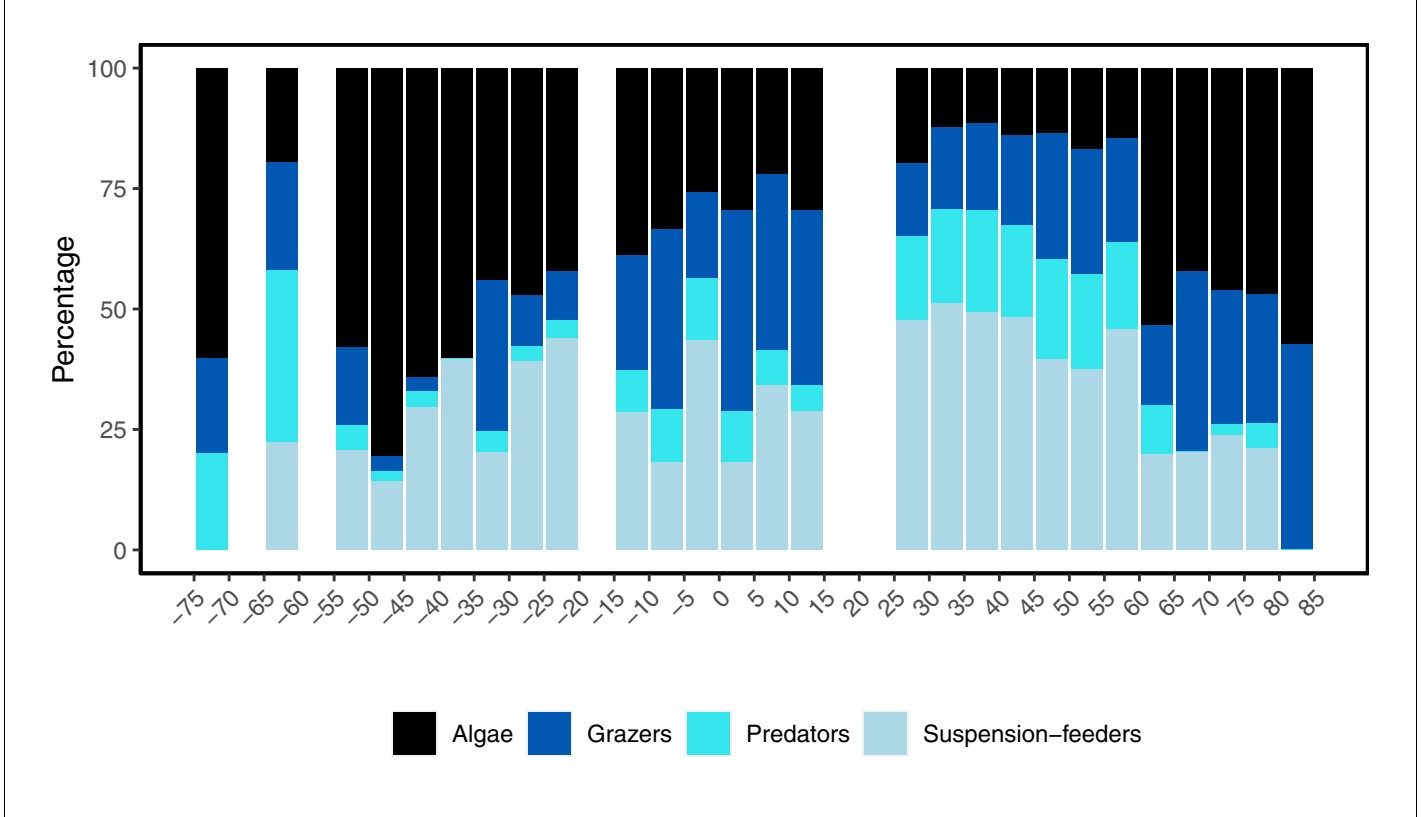

**Figure 5.** Latitudinal variation in the relative number of species from four functional groups in the rocky intertidal between 74.8°S and 80.5°N, where negative values denote southern hemisphere latitudes. No data were available within the 5° latitudinal bands where bars are missing.

(*Figure 5*). The relative proportion of algal species increased with latitude, with a small decline at sites between 60 and 65°S (*Figure 5*). In the northern hemisphere, predators were abundant at low- and mid-latitudes, accounting on average for 18.2% (±5.8 s.d.) of the populations from 0 to 60°N, before declining to 4.7% (±4.1 s.d.) at high latitudes between 61 and 80°N. The number of grazers decreased markedly from around 15°N, but stabilized at 25.4% (±6.4 s.d.) across 25°–80°N (*Figure 5*). Suspension feeders were the dominant functional group on shores in the northern hemisphere, especially between 25 and 60°N, but decreased to 21.4% (±1.8 s.d.) at high latitudes between 60 and 80°N (*Figure 5*). Algal species was the most dominant group at high latitudes (*Figure 5*). In both hemispheres, the assemblage composition at the highest latitudes was substantially different from elsewhere; on these high polar shores, suspension feeders were absent and only a few predators were present, while grazers and algae dominated.

## Discussion

### Latitudinal α-diversity gradients

Latitudinal diversity gradients have been recognized for centuries, and the generality of this phenomenon has been investigated ever since. Recent state-of-the-art meta-analyses have provided support for latitudinal gradients in various marine and terrestrial ecosystems (*Hillebrand, 2004*; *Kinlock et al., 2018*), and latitudinal gradients in assemblage composition and biological interactions have also been identified (*Qian and Ricklefs, 2007*; *Schemske et al., 2009*; *Harper and Peck, 2016*). Although latitudinal diversity gradients are widely quoted as one of the basic laws in ecology (*Lomolino, 2004*), studies investigating these patterns on a global scale have largely overlooked intertidal ecosystems, despite their global range and importance as unique environments. An intertidal study in the north eastern Pacific Ocean has reported no latitudinal gradients in species richness (*Blanchette et al., 2008*). It has been suggested that the latitudinal diversity gradient is

biogeographically structured (*Roy et al., 1994*), and since the above investigations were conducted in a temperate biogeographic region, diversity gradients might not occur. For example, while gastropod richness is relatively stable in the temperate region, a steep rise in richness occurs in the tropical region (*Fenberg and Rivadeneira, 2019*). However, the few published global-scale intertidal studies have revealed contradictory latitudinal patterns with richness of macroalgae (*Konar et al., 2010*) and small echinoderms peaking at higher latitudes, while large echinoderms peaking at low latitudes (*Iken et al., 2010*). Consequently, no latitudinal trend in intertidal diversity was demonstrated across 13 marine regions, as overall assemblage richness was similar everywhere despite strong species-specific patterns (*Cruz-Motta et al., 2010*). While the studies above benefited from analyzing data collected using a standardized protocol across latitudes, they were limited by only including a smaller number of sampling sites (~70 sites), which may have restricted their capacity to demonstrate large-scale trends, or to separate these from locally entrained patterns. Compared to these studies, we expanded the spatial resolution sixfold and present α-diversity data from 433 rocky shore sites across 155° of latitude, yet there was still no support for the latitudinal α-diversity gradient hypothesis. Latitudinal patterns were, however, markedly different between hemispheres, with a mid-latitude peak in the north and a non-significant unimodal trend in the south. We also demonstrated that different functional groups had different patterns (see later discussion).

Northern hemisphere α-diversity displayed a distinct mid-latitude peak, a pattern also reported for open water biodiversity (*Chaudhary et al., 2016*). The cause behind this remains elusive but hypotheses include a mid-dominance effect where high- and low-latitude species ranges overlap (*Powell et al., 2012*), and that high temperatures near the equator, and low temperatures and ice cover at the poles, may limit diversity outside of the temperate zone. However, we did not see this pattern in the temperate zone in the southern hemisphere, and we show that water temperatures do not control intertidal α-diversity. Instead, the open water mid-latitudinal richness peak has recently been suggested to be an artifact of limited sampling in less studied tropical regions (*Menegotto and Rangel, 2018*). This asymmetric sampling holds true for the northern hemisphere rocky intertidal ecosystem as well. While we show data from 11 sites at latitudes above 70° (south and north combined), 44% (n = 193) of all studies were from mid-latitude rocky shores in Europe and North America between 24 and 60°N. Data from low-latitude regions is furthermore limited because rocky shorelines are scarce in the tropics (*Fenberg and Rivadeneira, 2019*), and the majority of tropical studies have focused on specific taxonomic groups, and not assemblage-wide patterns (e.g., *Hartnoll, 1976*; *Chelazzi and Vannini, 1980*; *Mille-Pagaza et al., 2002*; *Konar et al., 2010*; *Flores-Rodríguez et al., 2014*; *Rivadeneira et al., 2015*; *Gbedemah, 2017*; *Hendrickx et al., 2019*, but see *Lourenço et al., 2020*). Thus, mid-latitudinal rocky shores, especially in the northern hemisphere, are among the most sampled globally, and future tropical collections may change understanding of biodiversity patterns. This disproportionate sampling pattern is also present in other marine habitats, where sampling intensity is highest at mid-latitudes in the northern hemisphere (*Menegotto and Rangel, 2018*). Polar data were also limited in the southern hemisphere where species diversity was relatively constant across latitudes with a non-significant dip at the highest latitude. A factor that powerfully affects intertidal diversity in the high polar regions is that most intertidal areas are encased in ice for large parts, if not all of the year, which reduces intertidal biodiversity to zero in these areas. Thus, the Arctic permafrost coastline represents around 34% of the world's coastline (*Lantuit et al., 2012*), and Antarctica accounts for around 2.7% (45,317 km). However, in the Antarctic only around 12% of that is ice-free in summer (5468 km) and at both poles much less, if any, is ice-free in winter (*Peck, 2018*). This lack of ice-free intertidal areas in high polar sites restricts the capacity for communities to establish, and limits the development of macroalgal species (*Miller and Pearse, 1991*; *Mystikou et al., 2014*). However, high polar shores are seldom studied (*Thyrring et al., 2021*), and in this study high-latitude Antarctic shorelines were only represented by a single study at a single site, thus this depression in number of species should be interpreted with caution. In ice-free areas, and at lower Antarctic latitudes, previous biodiversity studies have shown intertidal species richness to increase with latitude from the southern Atlantic ocean to the Antarctic Peninsula (*Waller, 2008*; *Griffiths and Waller, 2016*). The situation is clearly complex and, in general, more information on the influence of geographic sampling biases across latitudes are, together with more data from more sites, especially the polar and tropical regions, required to understand and model large-scale biodiversity patterns.

Latitudinal diversity gradients have been identified in subtidal habitats (e.g., *Roy et al., 2000*; *Edgar et al., 2017*), and the contrasting patterns in biodiversity gradients between intertidal and subtidal ecosystems may originate from differences in the factors controlling α-diversity. Subtidal benthic habitats experience relatively homogenous environmental conditions across small-to-moderate spatial scales, and large-scale latitudinal diversity gradients are therefore less impacted by local factors and reflect larger-scale changes in environmental conditions. For instance, decreasing water temperatures profoundly affect physiological performance of marine ectotherms (e.g., growth, reproduction and activity, muscle performance) and special adaptations are required to live in cold waters (*Peck, 2016*; *Peck, 2018*). Latitudinal biodiversity gradients in ectotherms have, therefore, been attributed to changes in water temperatures in subtidal habitats (*Roy et al., 2000*; *Edgar et al., 2017*). Our data run contrary to this and indicate that organisms inhabiting the intertidal do not conform to classic LDG patterns, or any of the correlated environmental drivers (e.g., water temperature). Instead, we demonstrate significant effects of local-scale variation in salinity, wave exposure, ice scour, and biogenic habitats (i.e., macroalgal canopy cover), indicating that α-diversity is determined by small-scale processes through biological interactions and overlapping environmental gradients. Indeed, the importance of microhabitats; biological interactions; and local physical parameters on intertidal community assemblage, density, and diversity has been studied extensively on rocky intertidal shores (*Paine, 1974*; *Archambault and Bourget, 1996*; *Coleman et al., 2006*; *Scrosati and Heaven, 2007*; *Piazzi et al., 2018*; *Sejr et al., 2021*), and high spatial variable in intertidal community coverage and structure between closely adjacent areas are well known (*Underwood and Chapman, 1996*; *Benedetti-Cecchi and Cinelli, 1997*; *Sejr et al., 2021*). For example, algal canopies provide food, predation refugium, and a moist understory environment where species sensitive to desiccation stress can survive during low tides (*Dayton, 1975*), thus the understory community may be rich and distinct from the surrounding open shoreline (*Benedetti-Cecchi et al., 2001*; *Bulleri et al., 2002*; *Watt and Scrosati, 2013*). Canopies and surface topography also shelter organisms from extreme air temperatures, thereby increasing local coverage and diversity, and facilitating recolonization of the substrate by secondary species (*Blanchard and Bourget, 1999*; *Watt and Scrosati, 2013*; *Ørberg et al., 2018*). Moreover, wave exposure decreases diversity, especially at high latitude where floating ice is pushed on the shore by waves, amplifying the impacts of ice scour on exposed coastlines (*Scrosati and Heaven, 2007*). Even wave splashing and low water timing (i.e., low tides occurring in the tropics during the hottest time of the day are more harmful than low tides in the middle of the night [*Helmuth et al., 2002*]) have been shown to be important. All these factors interact and change in non-latitudinally related patterns, creating highly localized unique environmental conditions that, in combination, functionally override latitudinal stress or energy gradients. Thus, although the strength and direction of environmental stressors change across latitudes (i.e., from ice scour and extreme sub-zero temperature in polar regions to desiccation and acute heat stress in the tropics), the combined stress experienced by resident populations may roughly balance out across most latitudes and produce no clear gradient. In a recent study in South America, α-diversity generally did not conform to latitudinal diversity gradient predictions for either the Atlantic or Pacific coast (*Cruz-Motta et al., 2020*), and we demonstrate similar patterns on a global scale. A conclusion from this research is that instead of focusing on large-scale drivers that implicitly assumed patterns and mechanisms are scale invariant, focus should be on how scales relevant to the organisms affect distribution patterns, as these factors outweigh latitudinal drivers of biodiversity. Notably, intertidal γ-diversity, which may in addition be controlled by regional drivers (e.g., upwelling, ocean currents) may display different latitudinal patterns than described here (*Cruz-Motta et al., 2020*), but global γ-diversity patterns remain to be explored in depth. In polar regions where glaciers or a persistent ice foot covers the intertidal, no communities are present (*Dayton, 1990*; *Peck, 2018*), and where the intertidal is characterized by a seasonal ice-foot formation, colonization is only possible during spring and summer (*Barnes, 1999*), although in these areas it is possible that a few species might persist in highly saline brine pools in winter (*Clarke and Beaumont, 2020*). Any future reduction of glaciers, ice shelves, and ice foots in these regions will therefore permit organisms to colonize the intertidal zone (*Węsławski et al., 2011*; *Kennicutt et al., 2014*; *Kennicutt et al., 2015*; *Kennicutt et al., 2019*). This will result in increased biomass and α-diversity as is evident from the high Arctic (*Weslawski et al., 2010*; *Kortsch et al., 2012*; *Thyrring et al., 2021*). Given that most studies fail to consider the aspects discussed above, and that data available from tropical and polar regions are sparse or absent, the relative impacts of

multiple stressors ranging from local scale to large scale cannot be explored in depth for coastal bio-diversity. However, our data show that latitudinal patterns in intertidal biodiversity differ significantly from subtidal studies. They also highlight the need for re-evaluating conservation efforts and climate change predictions for this unique global ecosystem.

## Latitudinal gradients in functional groups

In contrast to overall α-diversity, we identified strong latitudinal patterns in assemblage composition. The relative dominance of the four functional groups changed with latitude, with predators display-ing the strongest gradient. In the northern hemisphere, the number of predators decline at latitudes above 60°N, supporting local-scale studies showing that although intertidal predators, such as crabs and starfish, live on mid-latitude intertidal shores (*Paine, 1974*; *Jenkins et al., 2008*), they are miss-ing or rare on exposed Arctic shorelines (*Thorson, 1934*; *Weslawski et al., 2010*; *Høgslund et al., 2014*). A latitudinal reduction in the number of predators is consistent with the hypothesis of a gen-eral reduced level of predation in the polar regions (*Aronson et al., 2007*; *Schemske et al., 2009*; *Peck, 2018*), and similar latitudinal predation clines are found in subtidal benthic habitats (*Taylor and Taylor, 1977*; *Harper and Peck, 2016*). Several explanations have been proposed for this latitudinal decline, including the lack of durophagous predators in Antarctica, which has been attributed to the increased solubility of calcium carbonate ($CaCO_3$) with latitude, increasing the cost of shell production (*Vermeij, 1987*; *Watson et al., 2017*). The power muscles generation is also strongly affected by temperature and has been proposed as another reason why durophages are absent in high polar regions (*Aronson et al., 2007*; *Peck, 2018*), but there is still no full consensus on the processes behind these patterns, and the mechanisms explaining predation gradients remain debated. Predators are important for organizing food webs and community composition. For exam-ple, the habitat-forming mussel *Mytilus californianus* expanded and excluded more than 25 species after removal of its main predator, the starfish Pisaster, in Western North America (*Paine, 1974*), and in the absence of predators, a tropical seagrass community can support 10 times more species than with predators present (*Freestone et al., 2011*). Thus, the low proportion of predators at high latitudes indicates that the importance of top-down biological interactions in assemblages decreases with latitude in marine ecosystems, and that species composition in polar regions may primarily be controlled by the physical environment (*Barnes, 2002*; *Schemske et al., 2009*; *Høgslund et al., 2014*; *Peck, 2018*). However, it should be noted that we only studied benthic intertidal predators collected during low tides. Therefore, we cannot estimate impacts of mobile predators (e.g., fish and birds) or the predation pressure from epibenthic species during submersion of the shoreline, which can be significant (*Bertness et al., 1981*; *Ellis et al., 2007*).

Algae formed the most abundant functional group at high latitudes, consisting of both encrusting species surviving in surface depressions, protected from ice scour and extreme temperatures, and larger macrophytes. Some macrophytes found in polar regions are able to vegetatively regenerate tissue after substantial ice damage to fronds and fastholds, and can survive extreme sub-zero air temperatures during emersion (*Kiirikki and Ruuskanen, 1996*). Indeed air temperature is the prime stressor in polar intertidal systems (*Peck et al., 2006*; *Thyrring et al., 2020*), and microhabitats cre-ated by macroalgal fronds are important on intertidal shores across latitudes as they provide protec-tion that produces microclimates permitting survival during temperature extremes. The highest latitude shores contain fewer  visually obvious species (*Waller, 2008*), but coralline algae and encrusting species are capable of surviving extreme temperatures in microhabitats, and can survive the winter underneath the intertidal ice foot, likely in protected air pockets or saline brine tidepools reached occasionally and/or tidally by water (*Thyrring et al., 2017*; *Clarke and Beaumont, 2020*).

In this study, grazers occurred across all latitudes. Grazing can have powerful effects on algal bio-mass and distribution at low- to mid-latitudes (*Jenkins et al., 2008*; *Hawkins et al., 2019*), and graz-ing is an important driver of microalgal assemblage structure on King George Island in Antarctica (*Zacher et al., 2007*). Contrary to this, no patellid limpet has been reported from south Greenland, suggesting grazing is of little importance in this sub-Arctic region (*Høgslund et al., 2014*). Canopy-forming macroalgae are key structuring species at high latitudes (*Ørberg et al., 2018*), yet data on grazing impacts in polar intertidal habitats are currently too limited to make conclusions on the eco-logical implications. Indeed, more quantitative data are needed to fully understand the ecological implications of temporal and spatial changes in functional diversity, but overall our research

demonstrates that small-scale local environmental factors dictate the biodiversity patterns observed and overwhelm any large-scale patterns investigated.

## Materials and methods

### Data collection

In 2019 we searched the literature using Web of Science, Google Scholar, and Scopus for publications devoted to rocky intertidal local species richness, community composition, assemblage composition or α-diversity across all latitudes from the Antarctic to Arctic. Reports were considered valid when authors presented assemblage-wide data on 'α-diversity', 'richness', or 'number of species' in terms of species lists, tables, or graphs. Specifically, we excluded studies that focused exclusively on algae (e.g. *Konar et al., 2010*; *Gbedemah, 2017*), invertebrates (e.g. *Hartnoll, 1976*; *Mille-Pagaza et al., 2002*), or individual taxonomic groups. Full species lists were extracted whenever possible, and WebPlotDigitizer was used to extract graphic data (*Drevon et al., 2017*). Additional data were obtained from the Multi-Agency Rocky Intertidal Network (pacificrockyintertidal.org), the Alaska Ocean Observing System (aoos.org), and the South American Research Group on Coastal Ecosystems (SARCE) via the Ocean Biogeographic Information System (obis.org) website. The supplementary Appendix 1 – data sources provides a complete literature list and is available from the Zenodo repository (*Thyrring and Peck, 2021*).

In any of the sources used in this study to abstract data, the fraction of species sampled at a given site (inventory completeness) depends on the sampling effort by the collector and local conditions. There are statistical methods to minimize the influence of sampling biases, such as the jackknife estimator of species richness (*Smith and Pontius, 2006*), which has been developed to estimate regional γ-diversity from subsamples and abundance (*Cruz-Motta et al., 2010 Chao and Chiu, 2016*). However, we focused on α-diversity obtained from published records, tables, and figures, and these data were often presented as a single value without supplementary information on subsamples or abundances. Thus, we were unable to verify and compare sampling efforts within sites, but instead we collected data from 433 sites to give very large spatial resolution. To further minimize sampling bias among sites, we only considered studies where all species were collected according to described proscribed set of methodologies.

We validated taxonomic status by checking all the names in the World Register of Marine Species (WoRMS; http://www.marinespecies.org). Whenever species lists were available, we allocated species to functional groups (*Figure 1*). Species were allocated to one of four functional groups based on food acquisition; algae (all primary producers), grazers (including scrapers), predators (including scavengers), and suspension feeders. Finally, distribution of algae, grazers, predators, and suspension feeders were analyzed and presented as the relative percentage .

### Statistical analysis

The full dataset included latitude, longitude, country, and five latitudinal-scale oceanographic drivers (chlorophyll *a*, nitrate, phosphate, sea surface temperature, and silicate) available from NASA Earth Observations (NEO; https://neo.sci.gsfc.nasa.gov) and US National Ocean and Atmospheric Administration (NOAA) World Ocean Atlas (https://www.nodc.noaa.gov) as 1° latitudinal × 1° longitudinal cells. For each of the 433 α-diversity sites, we extracted values of chlorophyll *a*, nitrate, phosphate, sea surface temperature, and silicate from the 1° cell (latitudinal × longitude) in which the site was found. If data was unavailable from the specific cell, we used the value from the nearest available cell within the same latitudinal band (data was unavailable from the specific 1° cell for 5% of the sites).

Relationships between co-variates were assessed using Pearson's correlation coefficients (*Zuur et al., 2010*). This revealed collinearity between nitrate and phosphate (Pearson $r = 0.94$), silicate and temperature (Pearson $r = 0.73$), and oxygen and temperature (Pearson $r = 0.99$). Based on the correlation coefficients and the variance inflation factor (VIF; threshold <10) we excluded nitrate, oxygen, and silicate from models to avoid inference from correlation between covariates (*Montgomery and Peck, 1992*; *Zuur et al., 2010*) The final models therefore included latitude, chlorophyll *a,* sea surface temperature, and phosphate. Latitudinal α-diversity patterns in the northern and southern hemispheres were analyzed using R (*R Development Core Team, 2019*). Initial data

exploration revealed a non-linear residual pattern in the northern hemisphere, and we therefore analyzed this data using a GAMM with a negative binomial distribution ($\theta = 9$; Link function = log) from the gamm4 R package (*Wood and Scheipl, 2020*). The  smoother for latitude was fitted with a cubic spline regression (cs), and the optimal amount of smoothing was selected using cross-validation (*Zuur et al., 2014*). The southern hemisphere $\alpha$-diversity pattern was analyzed using a GLMM with a negative binomial distribution ($\theta = 15.58$; Link function = log). In both models, negative binomial distributions were fitted to ensure acceptable residual patterns and avoid over-dispersion. We included the factor 'country' as a random intercept. The final models were validated by plotting residuals versus fitted values, versus each covariate in the model, and versus each covariate not in the model (*Zuur et al., 2014*; *Zuur and Ieno, 2016*). Bubble plots, and variograms from the R package *gstat* (*Gräler et al., 2016*), were used to inspect model residuals for spatial correlations, and showed no indication of spatial autocorrelation in either model.

To investigate the general influence of local environmental stressors (stressors relevant to species at site-level), data on four local-scale environmental drivers (ice scour, macroalgal cover, salinity, and wave exposure) were extracted from the published literature (see Materials and method, described above) when available through text, tables, or direct author correspondence. We focused on 'macroalgal cover' of canopy-forming algae (i.e., non-canopy algae were excluded from this dataset) because canopies create understory living space and a predation refugia that increase $\alpha$-diversity (*Benedetti-Cecchi et al., 2001*; *Bulleri et al., 2002*; *Watt and Scrosati, 2013*). This data extraction resulted in four individual datasets, which were analyzed separately as most papers only investigated one of the oceanographic drivers; effects of salinity and wave exposure were separately analyzed using fitted standardized input variables (scale from 0 to 1) to estimate effect sizes of data originating from different scales, depending on the literature source (*Schielzeth, 2010*). We analyzed salinity effects using a generalized linear model (GLM) with a Poisson distribution, while ice scour and macroalgal cover effects were analyzed using a GLM with a negative binomial distribution because $\alpha$-diversity was characterized by over-dispersion (determined after visual inspection of residual patterns), but showed no zero-inflation (*Hilbe, 2011*). Wave exposure effects were initially analyzed using a GLM, but model residuals showed a non-linear pattern, and the data were re-analyzed using a generalized additive model (GAM) with a negative binomial distribution to ensure acceptable residual patterns (*Zuur, 2012*). All models were finally validated by inspection of standardized residual patterns plotted against fitted values. Effect sizes for GLM models were estimated using bias-corrected parametric bootstrap methods (10,000 iterations) (*Zuur and Ieno, 2016*), and the effect of the variable was considered significant when 95% CI did not overlap zero.

## Acknowledgements

The authors thank Brenda Konar, Takashi Noda, Tan Koh Siang, Robin Elahi, Valdivia Nelson, and Megan Dethier for providing intertidal data. We gratefully acknowledge the Valdivia Nelson's projects FONDECYT (grant #1141037) and CONICYT-PIA (ANILLO; grant #ART1101) for providing data from Chile and Antarctic, respectively. This study further utilized data collected by the South American Research Group on Coastal Ecosystems (SARCE) sponsored by TOTAL foundation and data collected by the Multi-Agency Rocky Intertidal Network (MARINe): a long-term ecological consortium funded and supported by many groups. Please visit pacificrockyintertidal.org for a complete list of the MARINe partners responsible for monitoring and funding these data. Data management has been primarily supported by BOEM (Bureau of Ocean Energy Management), NPS (National Parks Service), The David and Lucile Packard Foundation, and United States Navy. This work is a contribution to 'The future of Arctic biodiversity in a climate change era' project.

## Additional information

### Funding

| Funder | Grant reference number | Author |
| --- | --- | --- |
| Danmarks Frie Forskningsfond | 7027-00060B | Jakob Thyrring |
| Natural Environment Research Council | | Lloyd S Peck |

| | | | |
|---|---|---|---|
| H2020 Marie Skłodowska-Curie Actions | 797387 | | Jakob Thyrring |

The funders had no role in study design, data collection and interpretation, or the decision to submit the work for publication.

### Author contributions

Jakob Thyrring, Conceptualization, Data curation, Formal analysis, Funding acquisition, Validation, Investigation, Visualization, Methodology, Writing - original draft, Project administration, Writing - review and editing; Lloyd S Peck, Conceptualization, Supervision, Investigation, Visualization, Writing - review and editing

### Author ORCIDs

Jakob Thyrring (iD) https://orcid.org/0000-0002-1029-3105

### Decision letter and Author response

Decision letter https://doi.org/10.7554/eLife.64541.sa1
Author response https://doi.org/10.7554/eLife.64541.sa2

## Additional files

### Supplementary files

- Supplementary file 1. Latitudinal distribution of the intertidal study sites.

- Transparent reporting form

### Data availability

Data used in this study are freely available from the Multi-Agency Rocky Intertidal Network (MARINe; pacificrockyintertidal.org), the Ocean Biogeographic Information System (OBIS; obis.org) and the Alaska Ocean Observing System (AOOS; https://gulf-of-alaska.portal.aoos.org). Publications obtained from the literature search can be found on the Zenodo data repository (Thyrring & Peck, 2021).

The following dataset was generated:

| Author(s) | Year | Dataset title | Dataset URL | Database and Identifier |
|---|---|---|---|---|
| Thyrring J, Peck LS | 2021 | Literature and data for "Global gradients in intertidal species richness and functional groups" | https://zenodo.org/record/3942062#.YFRg9a-wmUk | Zenodo, 10.5281/zenodo.3942061 |

The following previously published dataset was used:

| Author(s) | Year | Dataset title | Dataset URL | Database and Identifier |
|---|---|---|---|---|
| Miloslavich P, Cruz-Motta JJ, Klein E, Caputo M, Castro EG, Herrera C, Hernández A, Carranza A, Gobin J, Romero L, Navarrete S, Sepúlveda R, Macaya E, Londoño-Cruz E, Mora C, da Rocha R, Lotufo T, Krull M, Nunes JLS, Barros F, Pellizari F, Flores A, Gutiérrez J, Bigatti G, | 2019 | Building a marine biodiversity observation network: an example from South America. | https://obis.org/dataset/81db945e-d319-4237-b49e-cd3e78937393 | Ocean Biodiversity Information System, sarce_rockyshores |

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
