## [Decision Letter]

**Acceptance summary:**

This study addresses an important topic of broad ecological interest and provides important insights into the role of local-scale processes in shaping patterns of species diversity, in order to (i) assess if there is a global latitudinal diversity gradient (using α diversity) of rocky shore organisms and its functional groups and, (ii) whether there are any large scale or local environmental predictors of richness patterns. The strength of this paper is the global coverage of studies analyzed, showing for the first time that rocky shore richness does not appear to peak in the tropics – in contrast to many other studies of marine and terrestrial systems.

**Decision letter after peer review:**

[Editors’ note: the authors submitted for reconsideration following the decision after peer review. What follows is the decision letter after the first round of review.]

Thank you for submitting your work entitled "Global gradients in intertidal species richness and functional diversity" for consideration by *eLife*. Your article has been reviewed by 2 peer reviewers, and the evaluation has been overseen by a Reviewing Editor and a Senior Editor. The following individual involved in the review of your submission has agreed to reveal their identity: Phillip Fenberg (Reviewer #2).

Our decision has been reached after consultation between the reviewers. Based on these discussions and the individual reviews below, we regret to inform you that your work will not be considered further for publication in *eLife*.

Your manuscript asks important questions related to the global distribution of intertidal marine biodiversity. Both reviewers and editors agreed that the global scope of the manuscript was interesting and that it could be a significant addition to the literature. Nonetheless, one of the external reviewers found important gaps in the statistical treatment of the data, and both external reviewers identified gaps in the literature review that can potentially affect your dataset and the outcomes arising from its analysis. We think however that a new improved version (addressing comments below) could be resubmitted to *eLife*. Please note that it would be treated as a new submission, but that we would try to recruit the same outside experts for evaluation.

Reviewer #1:

This manuscript tests for consistent latitudinal gradients in species richness, including in the species richness of some specified functional groups. The question is interesting and important, and has not been investigated extensively in rocky shore systems as far as I can tell (but see below). The authors' approach is to extract raw richness values (counts of species) from the literature, and then test for relationships between richness and a range of explanatory variables including latitude, using GLMs, GLMMs, and GAMs. I am not certain, but I think only univariate relationships are fitted to the data. The northern and southern hemispheres are tested separately as well as together, but different coastlines (e.g., E Pacific vs W Pacific vs W Atlantic) appear to be lumped together for analysis. Spatial structure (I think) is accounted for by establishing 5-degree "bins" and then putting random effects on those bins. I don't know if sites in the same latitudinal bin but different parts of the world (e.g., Chile vs Australia) are put in the same bin or different bins, but I think they are the same. Some recommendations I have for improvement of the study are related to the following:

1. One of the paper's stated aims is to study functional diversity. However, functional diversity (i.e., the breadth of functional space occupied by a biota, or the number of distinct functions represented by a biota) is not actually investigated in the paper. Rather, the authors analyze overall species richness, and then analyze species richness separately for a few different functional groups.

2. One previous study, by Cruz-Motta et al. 2010, is cited by the manuscript but its approach and findings are not discussed at all, which is surprising since the underlying questions are very similar. Another of which I'm aware of is Rivadeneira et al. (2015, DOI 10.1111/geb.12328) focus on gastropods only, but is also a very extensive study of the eastern Pacific latitudinal gradient. That paper is not cited in the current manuscript. The paper does not clearly explain how the current study improves upon those earlier studies or provides complementary insights, nor is there any explicit comparative discussion of the findings.

3. Presumably the species lists considered represent a very heterogeneous body of studies that differ substantially in their sampling effort and potentially their focal taxa (i.e., some studies may count all species in some unit of area, and others only species from certain taxa). Was there any consistency applied in study selection, such as: a set of focal taxa was identified by the authors, and a study was only included if it identified species comprehensively from each of those focal taxa? Sampling effort would no doubt have varied substantially as well, but there does not appear to be any standardization for sampling effort, either through the application of a richness estimator such as Chao1 or the jackknife, or rarefaction, or including some measure of sampling effort as a covariate in the analysis. Lastly, this is technically a meta-analysis, but meta-analytical methods (which would account for, say, differences in sampling variance among replicates in estimates of richness) do not appear to be employed. Given a large number of study sites, I doubt this would make a huge difference, relative to the other issues, so it is less of a concern to me than the other points.

4. There is a lot of important information missing from the description of the methods.

a. For example, how the random effects structure was imposed isn't entirely clear, although my best guess is that a random effect was placed on every 5 degrees latitudinal band. This isn't really an ideal way of coping with spatial autocorrelation, given that adjacent latitudinal bands are also likely to have some autocorrelation – and dealing with it is complicated by the fact that one of the explanatory variables is latitude (one way of modeling spatial autocorrelation is to use spatial coordinates as a trend surface, so in a sense, at least a specific form of the latitudinal component of spatial autocorrelation is in fact what is being tested for).

b. A "passion distribution" (I presume the authors mean Poisson?) was used to test the salinity model, but a negative binomial error distribution was used for the other model fits. This is confusing because the error distribution should depend on the behavior of the *response* variable (richness), not the explanatory variables. Moreover, the authors then talk about checking for normality and homoscedasticity which is confusing since a negative binomial distribution of residuals (presumably what was modeled) is neither normal nor homoscedastic.

c. It appears but was not clearly stated that only univariate relationships were tested. Why was there no model selection conducted investigating the potential for multiple variables to independently or interactively influence richness? Also, what are the statistical relationships among the potential explanatory variables? To what extent could variables that have statistically meaningful relationships with richness be coincidental, and due to the fact that they are spatially correlated with other variables that play a more causal role?

d. There are references in the text to global versus "regional" analyses, but what exactly these regions are isn't clear. Is it just northern and southern hemisphere data analyzed separately?

My suggestion for addressing the methods issues would be a more comprehensive presentation (or at least investigation and detailed description) of the model fit diagnostics, such as spatial correlograms of both residuals and random effects estimates for the different model fits. Structure in either would indicate something important missing. I would encourage the authors to revisit model diagnostics for models with non-normal error structures. Especially for things like the negative binomial, interpreting standard model diagnostic plots is extremely difficult and non-intuitive, and I'm concerned from the text about how much the authors dug into this. If the authors aren't sure where to start here, one option is to simulate data from a fitted model, and compare the distribution of the residuals from the model fitted to the real data with what the residuals look like when data are simulated to conform exactly to model assumptions. R package DHARMa will do that. Obviously, though that's not the only way forward. If there's evidence of lack of fit, the authors might consider some models with multiple explanatory variables. I am also surprised the authors didn't consider a model accounting for the potential effects of different coastlines. Along those lines, one way to account for "region" is to include it as a categorical factor in the original analysis, rather than do a completely different analysis. For example, E Pacific, W Pacific, E Atlantic, W Atlantic, etc. Hemisphere effects could be considered in this way also. If not, I would want to see that there is no such structure in the residuals (e.g., if one color-coded residual or random-effects estimates according to the coastline, for example, they would not be grouped together).

5. My overall take on the Discussion was that the study did not permit very strong conclusions to be drawn – for instance, the authors conclude that there is a mid-latitude peak in richness in mid-latitudes in the northern hemisphere but not the southern, and they attribute this to ice scour in the far north and desiccation and temperature at the equator. However, the authors don't clearly link this causal attribution to their analyses – the temperature effect looks non-significant, and the ice scour effect small in magnitude, which does not seem to support the authors' conclusion. My broad recommendation for the Discussion is for the causal inferences to be linked more concretely to the results of the analyses. A more thorough set of analyses that addresses some of the concerns that I raised in the previous points might make this somewhat easier.

Reviewer #2:

I really enjoyed reading this paper as it is very much within my area of expertise. The largely held belief that the latitudinal diversity gradient is a common pattern held across all major ecosystems is mostly substantiated by the literature. However, as you correctly state in your paper, it is rarely studied at the global scale in rocky shore systems. If it has been, then it is usually done using a coarse scale (e.g. 5 latitudinal bins) that also includes shallow sub-tidal environments, and/or using species richness based on overlapping geographic ranges (but see Rivadeneira et al. 2015 along the entirety of the eastern Pacific…below). Your findings suggest that there is no clear latitudinal gradient using α diversity on a global scale. While this is true based on your existing dataset, I do have one major comment that should help in your revision of the article:

There is a clear sampling gap in tropical latitudes (based on your figure 1). While this is partly because the rocky shore is sparse within tropical latitudes (e.g. Fenberg and Rivadeneira 2019; Ecology Letters), you are missing some regions in your dataset that clearly have papers on the α diversity of tropical rocky shore ecosystems. For example, along the eastern Pacific coast, you have no data points along the mainland Pacific coast of Mexico. Please have a look at Figure 3b in Rivadeneira et al. 2015 Global Ecology and Biogeography. This paper (which I am a co-author on) is based on rocky shore gastropods using literature sources across the whole of the eastern Pacific coast. While there are fewer examples of sites within the tropics compared to the temperate regions, there is a clear increase in species richness within the tropical latitudes (and this is just for gastropods). This runs counter to your argument that there is no gradient in α diversity, so you could include it in your paper as a potential outlier. Related to the above: see the following papers: the Revillagigedo Islands in Mexico, in Mille-Pagaza et al. 2002 ("Abundancia y diversidad de los invertebrados litorales de isla Socorro,Archipiélago Revillagigedo, México") they find 161 species of marine invertebrates (across a few different sites). The paper is written in Spanish, so it may have escaped your search terms. Here are a couple of other papers showing high diversity along the pacific rocky coast of Mexico:

Hendrickx et al. 2019: "Moluscos litorales (Bivalvia, Gastropoda, Polyplacophora, Cephalopoda) de playas

rocosas de la región de Guaymas, golfo de California, México" find 113 species.

Flores-Rodríguez, P., et al. (2014) Mollusks of the Rocky Intertidal Zone at Three Sites in Oaxaca,

Mexico. Open Journal of Marine Science, 4, 326-337.

These are just a few examples of Mexican papers that you have missed (that seem to fit your search criteria).

You also have missed tropical papers along the coast of eastern Africa: Tanzania: Hartnoll Estuarine and Coastal Marine Science (1976)

Somalia: Chelazzi and Vannini (1979) "Zonation of Intertidal Molluscs on Rocky

Shores of Southern Somalia".

And it appears that you have missed papers from New South Wales (your example from Figure 1 appears to be in Queensland): Underwood 1981 "Structure of a rocky intertidal community in New South Wales: patters of vertical distribution and seasonal changes".

Benkendorff and A.R. Davis (2002): "Identifying hotspots of molluscan species richness on rocky intertidal reefs"

These are just a few examples of tropical papers that seem to fit your search criteria but are missing in your analysis. The point I am trying to make here is that while there may indeed be fewer papers in the tropical regions, there are definitely more to be found. While this may not change the overall results of your paper, I think you should adjust your search criteria as you are likely missing quite a few from tropical rocky shores. Some of the papers will be in different languages (especially the Spanish papers from Mexico and Latin America), but they should not be discounted in my opinion because some of them clearly show high levels of α diversity within the tropical latitudes. In conclusion, I really do like this paper and I feel like it will make a valuable contribution – but I feel like the dataset is incomplete and would benefit from a more thorough search for papers within tropical rocky shores.

[Editors’ note: further revisions were suggested prior to acceptance, as described below.]

Thank you for submitting your article "Global gradients in intertidal species richness and functional groups" for consideration by *eLife*. Your article has been reviewed by 2 peer reviewers, and the evaluation has been overseen by Detlef Weigel as the Senior and Reviewing Editor. The following individuals involved in review of your submission have agreed to reveal their identity: Phillip Fenberg (Reviewer #1); Lisandro Benedetti-Cecchi (Reviewer #2).

The reviewers have discussed the reviews with one another and the Reviewing Editor has drafted this decision to help you prepare a revised submission.

Summary:

This study addresses an important topic of broad ecological interest and provides important insights into the role of local-scale processes in shaping patterns of species diversity, aiming to (i) assess if there is a global latitudinal diversity gradient (using α diversity) of rocky shore organisms and its functional groups and, (ii) whether there are any large scale or local environmental predictors of richness patterns. The strength of this paper is the global coverage of studies analyzed, showing for the first time that rocky shore richness does not appear to peak in the tropics – in contrast to many other studies of marine and terrestrial systems. These outcomes are not specific for rocky intertidal systems, with an increasing number of studies showing that the search for global ecological patterns may be elusive. While sampling in the tropics and the polar regions is poor (acknowledged by the authors), this should be viewed as a call for further research in these regions – not as a weakness of the paper per se. There are also some reservations on how the analysis has been conducted, including the lack of standardization of sampling effort and other details (e.g., size of sampling units) to derive a comparable measure of diversity across sites.

Essential revisions:

The latitudinal gradient of diversity has been studied and confirmed in many aquatic and terrestrial habitats and species across the globe. In the vast majority of cases, richness increases towards the tropics. Using an impressive global dataset of latitudinal diversity gradients in 433 rocky intertidal assemblages of algae and invertebrates from the Arctic to the Antarctic, Thyrring and Peck show that rocky shore ecosystems may not follow this general pattern. The authors show that there is no clear latitudinal gradient for rocky shore organisms using alpha diversity - as posited by prevailing theories - although some functional groups exhibit contrasting patterns. Diversity within functional groups of predators, grazers and filter-feeders decreased towards the poles, whereas the opposite was observed for macroalgae. Correlation with environmental drivers highlighted the importance of local-scale processes in driving spatial patterns of diversity in rocky intertidal assemblages. The paper is well written and the many of the analyses are well done, but there is the concern, which the authors acknowledge, that sampling within tropical latitudes is sparse and needs to be carefully considered when interpreting the results of this paper.

1. The relevant data to standardize species richness may not be available from the primary literature. However, it should be possible to employ relevant standardization methods within the 5{degree sign} latitudinal bands in which the data have been aggregated. An analysis based on standardized data, at least for the more data-rich latitudinal bands, must be added.

2. Employ models that allow assessing unimodality, which is stated but untested. At the bare minimum, a quadratic relationship with latitude should be included in the GLMM. As implemented here, the GLMM employed to relate diversity to latitude can only detect linear trends, but not unimodal patterns and the mid-latitude peak suggested by LOESS for the northern hemisphere. To provide a formal test for unimodality, models with or without a quadratic term could be contrasted using standard model comparison procedures. Alternatively, GAM could be used to evaluate nonlinear effects.

3. Clarify whether p-values are relevant or not. As is, it is confusing. For example, the legend of Table 1 mentions p-values, but these are not reported. Materials and methods indicate that 95% confidence intervals are used to take decisions on null hypotheses, suggesting that p-values are not used in the analysis (lines 436-439). Nevertheless, p-values are reported in Table 2.

4. Provide a rationale for distinguishing between canopy and other algal forms (the distinction is compelling, but it is not explained).

5. We like the conclusion on the importance of local-scale processes. This should be placed in the context of previous studies that have quantified patterns and processes at multiple scales reaching the same conclusion.

6. We could not access the data repository indicated in ref. 91, so we could not assess whether the analysis may have missed potentially relevant papers.

7. Provide the number of studies available for each band in an Appendix.

8. The analysis on macroalgae (e.g., Figure 5) distinguished between canopy and no-canopy algae. This is probably correct, but the rationale for this distinction has not been provided. I think some context is needed, especially to clarify the role of algal canopies in maintaining diversified understory assemblages.

9. The overall conclusion that more studies are needed to assess the magnitude and influence of physical and biotic drivers across multiple scales is important and appropriate. However, many studies have examined processes across scales in rocky intertidal systems (including canopy-removal and limpet-exclusion experiments) and many descriptive studies have quantified variation across multiple spatial scales, emphasizing the importance of small-scale variability in pattern of distribution, abundance and diversity of species on rocky shores. A more thorough discussion of this literature (e.g., Underwood & Chapman, Benedetti-Cecchi, Denny, Coleman, Martins, Fraschetti, etc) would be welcome.

10. In the abstract please say something about more sampling being needed in the tropics. Perhaps line 34.

11. Line 85: What do you mean by "inadequate estimates of intertidal areas"? Inadequate in what way? And in what sense are you talking about area?

12. Line 96: Replace "controlled" by "predicted".

13. Figure 2: where are the R2 values on this plot? How do you get an R2 from a LOESS fit…?

14. Salinity is a local and a regional variable in your analyses, please briefly explain why (Figure 3 and 4).

15. Figure 5: hard to see the box plots, consider making them a different colour or shade.

---

## [Author Response]

[Editors’ note: the authors resubmitted a revised version of the paper for consideration. What follows is the authors’ response to the first round of review.]

Reviewer #1:This manuscript tests for consistent latitudinal gradients in species richness, including in the species richness of some specified functional groups. The question is interesting and important, and has not been investigated extensively in rocky shore systems as far as I can tell (but see below). The authors' approach is to extract raw richness values (counts of species) from the literature, and then test for relationships between richness and a range of explanatory variables including latitude, using GLMs, GLMMs, and GAMs. I am not certain, but I think only univariate relationships are fitted to the data. The northern and southern hemispheres are tested separately as well as together, but different coastlines (e.g., E Pacific vs W Pacific vs W Atlantic) appear to be lumped together for analysis. Spatial structure (I think) is accounted for by establishing 5-degree "bins" and then putting random effects on those bins.I don't know if sites in the same latitudinal bin but different parts of the world (e.g., Chile vs Australia) are put in the same bin or different bins, but I think they are the same.

Yes, sites across all longitudes within in 5° latitudinal bin was put together to analysis the generality of rocky shores LDGs across all oceans. The method of including sites from all longitudes in this way follows previous investigations on LDG in both terrestrial and marine systems [1–6]. We have highlighted this method in the Materials and methods section:

“Sites across all longitudes was binned within corresponding 5° latitudinal bands” (Manuscript line 403-404).

Some recommendations I have for improvement of the study are related to the following:1. One of the paper's stated aims is to study functional diversity. However, functional diversity (i.e., the breadth of functional space occupied by a biota, or the number of distinct functions represented by a biota) is not actually investigated in the paper. Rather, the authors analyze overall species richness, and then analyze species richness separately for a few different functional groups.

We agree the term ‘functional diversity’ was inappropriate when we actually analyse patterns in various functional groups. We now refer to “functional groups” throughout the text, and have change the title accordingly:

“Global gradients in intertidal species richness and functional groups”

2. One previous study, by Cruz-Motta et al. 2010, is cited by the manuscript but its approach and findings are not discussed at all, which is surprising since the underlying questions are very similar. Another of which I'm aware of is Rivadeneira et al. (2015, DOI 10.1111/geb.12328) focus on gastropods only, but is also a very extensive study of the eastern Pacific latitudinal gradient. That paper is not cited in the current manuscript. The paper does not clearly explain how the current study improves upon those earlier studies or provides complementary insights, nor is there any explicit comparative discussion of the findings.

We agree, the original discussion did not reflect the relevant literature (such as, Cruz-Motta et al. (2010), Konar et al. (2010), Rivadeneira et al. (2015) [7–9]), or explained how we improved upon these. In the revised version we provide a comprehensive discussion about the current knowledge on global intertidal diversity gradients, and highlight how we expand on this and provide new information:

“Although latitudinal diversity gradients are widely quoted as one of the basic laws in ecology [43], studies investigating these patterns on a global scale have largely overlooked intertidal ecosystems, despite their global range and importance as unique environments. […] We also demonstrated different functional groups had different gradients (see later discussion).” (Manuscript lines 204-223).

We have also rephrased parts of the introduction, so it now more correctly describes the current knowledge on the contradicting intertidal gradients:

“Regional-scale intertidal studies have found richness gradients of gastropods along coastlines in the eastern Pacific Ocean [7,15]. However, no latitudinal diversity gradient of gastropods [16] or macroalgal [8] was found on a global-scale across oceans, and assembly-wide studies have found missing [9,11,12] or inverse [17] gradients.” (Manuscript lines 73-76).

3. Presumably the species lists considered represent a very heterogeneous body of studies that differ substantially in their sampling effort and potentially their focal taxa (i.e., some studies may count all species in some unit of area, and others only species from certain taxa). Was there any consistency applied in study selection, such as: a set of focal taxa was identified by the authors, and a study was only included if it identified species comprehensively from each of those focal taxa? Sampling effort would no doubt have varied substantially as well, but there does not appear to be any standardization for sampling effort, either through the application of a richness estimator such as Chao1 or the jackknife, or rarefaction, or including some measure of sampling effort as a covariate in the analysis.

Sampling efforts and inventory completeness are important drivers for the number of species found in any survey. Studies investigating regional richness, or γ-diversity, should, as the reviewer suggest, apply richness estimators, such as Chao1 [18] or jackknife [19]. These methods rely on subsamples and abundances for estimating regional richness. However, we analyse a-diversity from different sites using data obtained from text, figures or tables, which was often only available in the form of a single value. Thus, any supplement information on abundance, sampling protocol or sampling effort was often missing. We acknowledge this caveat, and in order to minimize data biases we collected data from 433 sites to maximize spatial resolution, and minimise the influence from any odd studies. To further minimise sampling bias among sites, we only considered studies where all species was collected, according to the described methodologies. Thus, we excluded studies that focused exclusively on algae, invertebrates or individual taxonomic groups (see also our response to Reviewer 2 below). We have added this information in the Material and Method section:

“In any of the sources used in this study to abstract data, the fraction of species sampled at a given site (inventory completeness) depends on the sampling effort by the collector and local conditions. […] To further minimize sampling bias among sites, we only considered studies where all species were collected according to described proscribed set of methodologies. (Manuscript lines 380-389).

Lastly, this is technically a meta-analysis, but meta-analytical methods (which would account for, say, differences in sampling variance among replicates in estimates of richness) do not appear to be employed. Given a large number of study sites, I doubt this would make a huge difference, relative to the other issues, so it is less of a concern to me than the other points.4. There is a lot of important information missing from the description of the methods.a. For example, how the random effects structure was imposed isn't entirely clear, although my best guess is that a random effect was placed on every 5 degrees latitudinal band. This isn't really an ideal way of coping with spatial autocorrelation, given that adjacent latitudinal bands are also likely to have some autocorrelation – and dealing with it is complicated by the fact that one of the explanatory variables is latitude (one way of modeling spatial autocorrelation is to use spatial coordinates as a trend surface, so in a sense, at least a specific form of the latitudinal component of spatial autocorrelation is in fact what is being tested for).

We agree the statistical analysis needed much revision, and we thank the reviewer for the constructive suggestions. Accordingly, we re-analysed the data and looked for spatial autocorrelation using Variograms and bubble plots. The data actually showed clear autocorrelation, and accordingly all GLMMs were fitted with a Matérn spatial covariance structure with longitudinal and latitudinal values as spatial variables. We still nested sites in 5° latitudinal bins and used these as a random factor to account for dependency among sites nested within bands. The final GLMMs were validated by plotting standardized residual patterns plotted against fitted values, and using residual diagnostic tools form the DHARMa R package [28]. Please see the fully revised Materials and methods section for more details.

b. A "passion distribution" (I presume the authors mean Poisson?) was used to test the salinity model, but a negative binomial error distribution was used for the other model fits. This is confusing because the error distribution should depend on the behavior of the response variable (richness), not the explanatory variables. Moreover, the authors then talk about checking for normality and homoscedasticity which is confusing since a negative binomial distribution of residuals (presumably what was modeled) is neither normal nor homoscedastic.

We apologies for the typo. Indeed, we meant a Poisson distribution, and this has been corrected in the text.

The local models (ice-scour, macroalgal cover, salinity and wave exposure) were analysed as 4 separate models because the available data varied among the four drivers. Consequently, we had four datasets with individual response variables (richness), and the error distribution was therefore different in each analysis to ensure valid models were used. We have clarified this in the Material and Method section to avoid any confusion:

“Because the available data varied among the four parameters, this data extraction resulted in four individual datasets, which were analysed separately as most papers only investigated one of oceanographic drivers” (Manuscript lines 422-424).

c. It appears but was not clearly stated that only univariate relationships were tested. Why was there no model selection conducted investigating the potential for multiple variables to independently or interactively influence richness? Also, what are the statistical relationships among the potential explanatory variables? To what extent could variables that have statistically meaningful relationships with richness be coincidental, and due to the fact that they are spatially correlated with other variables that play a more causal role?

We have clarified how the data exploitation process was conducted before the analysis. We tested for collinearity among environmental covaries, using a variance inflation factor (VIF) analysis, which showed no collinearity. Since environmental drivers may interact (e.g. temperature and salinity), the original full model included interactions among all covariates. However, there were no significant interactions, and we therefore excluded interactions in the final model [29]. Please see the substantially revised Materials and methods section.

d. There are references in the text to global versus "regional" analyses, but what exactly these regions are isn't clear. Is it just northern and southern hemisphere data analyzed separately?

The regional analyses were meant as an analysis focusing on small-scale drivers (e.g. salinity, wave exposure) that changes on a small spatial scale and among sites. We have rephrased the text, and now the “regional” analyses are called “local-scale” to symbolise its drivers relevant to specific sites. This has been changed throughout the text and we have expanded the describing in the Materials and Method section:

**“**To investigate the general influence of local environmental stressors (stressors relevant to species at site-level), data on four local-scale environmental drivers (ice-scour, macroalgal cover, salinity and wave exposure) were extracted from the published literature (see method described above) when available through text, tables or direct author correspondence” (Manuscript lines 419-422).

My suggestion for addressing the methods issues would be a more comprehensive presentation (or at least investigation and detailed description) of the model fit diagnostics, such as spatial correlograms of both residuals and random effects estimates for the different model fits. Structure in either would indicate something important missing. I would encourage the authors to revisit model diagnostics for models with non-normal error structures. Especially for things like the negative binomial, interpreting standard model diagnostic plots is extremely difficult and non-intuitive, and I'm concerned from the text about how much the authors dug into this. If the authors aren't sure where to start here, one option is to simulate data from a fitted model, and compare the distribution of the residuals from the model fitted to the real data with what the residuals look like when data are simulated to conform exactly to model assumptions. R package DHARMa will do that. Obviously, though that's not the only way forward. If there's evidence of lack of fit, the authors might consider some models with multiple explanatory variables. I am also surprised the authors didn't consider a model accounting for the potential effects of different coastlines. Along those lines, one way to account for "region" is to include it as a categorical factor in the original analysis, rather than do a completely different analysis. For example, E Pacific, W Pacific, E Atlantic, W Atlantic, etc. Hemisphere effects could be considered in this way also. If not, I would want to see that there is no such structure in the residuals (e.g., if one color-coded residual or random-effects estimates according to the coastline, for example, they would not be grouped together).5. My overall take on the Discussion was that the study did not permit very strong conclusions to be drawn – for instance, the authors conclude that there is a mid-latitude peak in richness in mid-latitudes in the northern hemisphere but not the southern, and they attribute this to ice scour in the far north and desiccation and temperature at the equator. However, the authors don't clearly link this causal attribution to their analyses – the temperature effect looks non-significant, and the ice scour effect small in magnitude, which does not seem to support the authors' conclusion. My broad recommendation for the Discussion is for the causal inferences to be linked more concretely to the results of the analyses. A more thorough set of analyses that addresses some of the concerns that I raised in the previous points might make this somewhat easier.

We have, as suggested, rewritten large parts of the discussion to better link our results and the discussion. We hope out revised discussion more clearly link our work and previous studies. Please see the revised Discussion section.

Reviewer #2:I really enjoyed reading this paper as it is very much within my area of expertise. The largely held belief that the latitudinal diversity gradient is a common pattern held across all major ecosystems is mostly substantiated by the literature. However, as you correctly state in your paper, it is rarely studied at the global scale in rocky shore systems. If it has been, then it is usually done using a coarse scale (e.g. 5 latitudinal bins) that also includes shallow sub-tidal environments, and/or using species richness based on overlapping geographic ranges (but see Rivadeneira et al. 2015 along the entirety of the eastern Pacific…below). Your findings suggest that there is no clear latitudinal gradient using α diversity on a global scale. While this is true based on your existing dataset, I do have one major comment that should help in your revision of the article:There is a clear sampling gap in tropical latitudes (based on your figure 1). While this is partly because the rocky shore is sparse within tropical latitudes (e.g. Fenberg and Rivadeneira 2019; Ecology Letters), you are missing some regions in your dataset that clearly have papers on the α diversity of tropical rocky shore ecosystems. For example, along the eastern Pacific coast, you have no data points along the mainland Pacific coast of Mexico. Please have a look at Figure 3b in Rivadeneira et al. 2015 Global Ecology and Biogeography. This paper (which I am a co-author on) is based on rocky shore gastropods using literature sources across the whole of the eastern Pacific coast. While there are fewer examples of sites within the tropics compared to the temperate regions, there is a clear increase in species richness within the tropical latitudes (and this is just for gastropods). This runs counter to your argument that there is no gradient in α diversity, so you could include it in your paper as a potential outlier. Related to the above: see the following papers: the Revillagigedo Islands in Mexico, in Mille-Pagaza et al. 2002 ("Abundancia y diversidad de los invertebrados litorales de isla Socorro,Archipiélago Revillagigedo, México") they find 161 species of marine invertebrates (across a few different sites). The paper is written in Spanish, so it may have escaped your search terms. Here are a couple of other papers showing high diversity along the pacific rocky coast of Mexico:Hendrickx et al. 2019: "Moluscos litorales (Bivalvia, Gastropoda, Polyplacophora, Cephalopoda) de playasrocosas de la región de Guaymas, golfo de California, México" find 113 species.Flores-Rodríguez, P., et al. (2014) Mollusks of the Rocky Intertidal Zone at Three Sites in Oaxaca,Mexico. Open Journal of Marine Science, 4, 326-337.These are just a few examples of Mexican papers that you have missed (that seem to fit your search criteria).You also have missed tropical papers along the coast of eastern Africa: Tanzania: Hartnoll Estuarine and Coastal Marine Science (1976)Somalia: Chelazzi and Vannini (1979) "Zonation of Intertidal Molluscs on RockyShores of Southern Somalia".And it appears that you have missed papers from New South Wales (your example from Figure 1 appears to be in Queensland): Underwood 1981 "Structure of a rocky intertidal community in New South Wales: patters of vertical distribution and seasonal changes".Benkendorff and A.R. Davis (2002): "Identifying hotspots of molluscan species richness on rocky intertidal reefs"These are just a few examples of tropical papers that seem to fit your search criteria but are missing in your analysis. The point I am trying to make here is that while there may indeed be fewer papers in the tropical regions, there are definitely more to be found. While this may not change the overall results of your paper, I think you should adjust your search criteria as you are likely missing quite a few from tropical rocky shores. Some of the papers will be in different languages (especially the Spanish papers from Mexico and Latin America), but they should not be discounted in my opinion because some of them clearly show high levels of α diversity within the tropical latitudes. In conclusion, I really do like this paper and I feel like it will make a valuable contribution – but I feel like the dataset is incomplete and would benefit from a more thorough search for papers within tropical rocky shores.

We agree that there is a rich literature on diversity from intertidal shores in various tropical regions, including the very nice examples listed above. Unfortunately, the majority of tropical studies focus on specific taxonomic groups, or they only investigate either the flora or fauna. This is also the case for the studies suggested above, specifically:

Hendrickx et al. 2019 – *invertebrates*

Flores-Rodríguez, P., et al. (2014) – *molluscs*

Rivadeneira et al. (2015) – *gastropods*

Fenberg and Rivadeneira (2019) – *gastropods*

Mille-Pagaza et al. (2002) –*invertebrates*

Benkendorff & Davis (2002) – *molluscs*

Chelazzi and Vannini (1979) – *invertebrates*

Hartnoll (1976) –*invertebrates*

Benkendorff & Davis (2002) – *molluscs*

We are grateful for the introduction to the paper by Underwood (1981), which data we now include (e.g. see figure 1). Unfortunate, the other papers are unsuitable for our study as we focus on assembly-wide a-diversity, and therefore cannot include papers on specific taxa or groups (i.e. only algae or animals). We clearly had not highlighted the selection criteria in the original method sections, and we have made substantial changes to the Material and Method section to avoid confusion: For example:

“Reports were considered valid when authors presented assembly-wide data on “a-diversity”, “richness” or “number of species” in terms of species lists, tables or graphs. Specifically, we excluded studies that focused exclusively on algae [26,49], invertebrates [50,51] or individual taxonomic groups [e.g. 24,25,51–53,87]. Full species lists were extracted whenever possible, and WebPlotDigitizer was used to extract graphic data [90].” (Manuscript lines 370-375).

References:

1. Malizia A, Blundo C, Carilla J, Osinaga Acosta O, Cuesta F, Duque A, et al. Elevation and latitude drives structure and tree species composition in Andean forests: Results from a largescale plot network. PLoS One. 2020;15: e0231553. doi:10.1371/journal.pone.0231553

2. Menegotto A, Rangel TF. Mapping knowledge gaps in marine diversity reveals a latitudinal gradient of missing species richness. Nat Commun. 2018;9: 4713. doi:10.1038/s41467-01807217-7

3. Chaudhary C, Saeedi H, Costello MJ. Bimodality of latitudinal gradients in marine species richness. Trends Ecol Evol. 2016;31: 670–676. doi:10.1016/j.tree.2016.06.001

4. Mateo RG, Broennimann O, Normand S, Petitpierre B, Araújo MB, Svenning J-C, et al. The mossy north: an inverse latitudinal diversity gradient in European bryophytes. Sci Rep. 2016;6: 25546. doi:10.1038/srep25546

5. Valdovinos C, Navarrete SA, Marquet PA. Mollusk species diversity in the Southeastern Pacific: why are there more species towards the pole? Ecography 2003;26: 139–144.

doi:10.1034/j.1600-0587.2003.03349.x

6. Roy K, Jablonski D, Valentine JW, Rosenberg G. Marine latitudinal diversity gradients: tests of causal hypotheses. Proc Natl Acad Sci U S A. 1998;95: 3699–702.

doi:10.1073/PNAS.95.7.3699

7. Rivadeneira MM, Alballay AH, Villafaña JA, Raimondi PT, Blanchette CA, Fenberg PB. Geographic patterns of diversification and the latitudinal gradient of richness of rocky intertidal gastropods: the ‘into the tropical museum’ hypothesis. Glob Ecol Biogeogr.

2015;24: 1149–1158. doi:10.1111/geb.12328

8. Konar B, Iken K, Cruz-Motta JJ, Benedetti-Cecchi L, Knowlton A, Pohle G, et al. Current patterns of macroalgal diversity and biomass in northern hemisphere rocky shores. Thrush S, editor. PLoS One. 2010;5: e13195. doi:10.1371/journal.pone.0013195

9. Cruz-Motta JJ, Miloslavich P, Palomo G, Iken K, Konar B, Pohle G, et al. Patterns of spatial variation of assemblages associated with intertidal rocky shores: a global perspective. PLoS One. 2010;5: e14354. doi:10.1371/journal.pone.0014354

10. Lomolino M V. Conservation biogeography. In: Lomolino M V., Heaney LR, editors. Frontiers of Biogeography: New Directions of the Geography of Nature. Sinauer Associates, Inc; 2004. pp. 293–296.

11. Fenberg PB, Menge BA, Raimondi PT, Rivadeneira MM. Biogeographic structure of the northeastern Pacific rocky intertidal: the role of upwelling and dispersal to drive patterns. Ecography 2015;38: 83–95. doi:10.1111/ecog.00880

12. Blanchette CA, Miner CM, Raimondi PT, Lohse D, Heady KEK, Broitman BR.

Biogeographical patterns of rocky intertidal communities along the Pacific coast of North

America. J Biogeogr. 2008;35: 1593–1607. doi: 10.1111/J.1365-2699.2008.01913.X

13. Schoch GC, Menge BA, Allison G, Kavanaugh M, Thompson SA, A. Wood S. Fifteen degrees of separation: Latitudinal gradients of rocky intertidal biota along the California Current. Limnol Oceanogr. 2006;51: 2564–2585. doi:10.4319/lo.2006.51.6.2564

14. Iken K, Konar B, Benedetti-Cecchi L, Cruz-Motta JJ, Knowlton A, Pohle G, et al. Largescale spatial distribution patterns of echinoderms in nearshore rocky habitats. PLoS One. 2010;5: e13845. doi:10.1371/journal.pone.0013845

15. Fenberg PB, Rivadeneira MM. On the importance of habitat continuity for delimiting biogeographic regions and shaping richness gradients. Ecol Lett. 2019;22: 664–673.

doi:10.1111/ele.13228

16. Miloslavich P, Cruz-Motta JJ, Klein E, Iken K, Weinberger V, Konar B, et al. Large-scale spatial distribution patterns of gastropod assemblages in rocky shores. PLoS One. 2013;8:

e71396. doi:10.1371/journal.pone.0071396

17. Griffiths HJ, Waller CL. The first comprehensive description of the biodiversity and biogeography of Antarctic and Sub-Antarctic intertidal communities. J Biogeogr. 2016;43: 1143–1155. doi:10.1111/jbi.12708

18. Yang W, Ma K, Kreft H. Geographical sampling bias in a large distributional database and its effects on species richness-environment models. J Biogeogr. 2013;40: 1415–1426.

doi:10.1111/jbi.12108

19. Smith CD, Pontius JS. Jackknife estimator of species richness with S-PLUS. J Stat Softw. 2006;15: 1–12. doi:10.18637/jss.v015.i03

20. Chao A, Chiu C-H. Nonparametric Estimation and Comparison of Species Richness. eLS.

Chichester, UK: John Wiley & Sons, Ltd; 2016. pp. 1–11.

doi:10.1002/9780470015902.a0026329

21. Gbedemah ST. Current patterns in intertidal macro-algal diversity and zonation of two sites on Ghana’s coast. J Oceanogr Mar Res. 2017;5: 159. doi:10.4172/2572-3103.1000159

22. Hartnoll RG. The ecology of some rocky shores in tropical East Africa. Estuar Coast Mar 1976;4: 1–21. doi:10.1016/0302-3524(76)90002-5

23. Mille-Pagaza S, Carrillo-Laguna J, Pérez-Chi A, Sánchez-Salazar ME. Abundancia y diversidad de los invertebrados litorales de isla Socorro, Archipiélago Revillagigedo, México. Rev Biol Trop. 2002;50: 97–105.

24. Flores-Rodríguez P, Flores-Garza R, García-Ibáñez S, Torreblanca-Ramírez C, GaleanaRebolledo L, Santiago-Cortes E. Mollusks of the rocky intertidal zone at three sites in Oaxaca, Mexico. Open J Mar Sci. 2014;04: 326–337. doi:10.4236/ojms.2014.44029

25. Chelazzi G, Vannini M. Zonation of intertidal molluscs on rocky shores of Southern Somalia. Estuar Coast Mar Sci. 1980;10: 569–583. doi:10.1016/S0302-3524(80)80076-4

26. Hendrickx ME, Salgado-Barragán J, Cordero-Ruiz M. Moluscos litorales (Bivalvia, Gastropoda, Polyplacophora, Cephalopoda) de playas rocosas de la región de Guaymas, golfo de California, México = Littoral mollusks (Bivalvia, Gastropoda, Polyplacophora, Cephalopoda) from rocky beaches in the area of Guaymas, Gulf of California, Mexico.

GeomareZoológica. 2019;

27. Benkendorff K, Davis AR. Identifying hotspots of molluscan species richness on rocky intertidal reefs. Biodivers Conserv. 2002;11: 1959–1973. doi:10.1023/A:1020886526259

28. Hartig F. DHARMa: Residual Diagnostics for Hierarchical (Multi-Level / Mixed) Regression Models. R package version 0.3.2.0. https://CRAN.R-project.org/package=DHARMa.

29. Ieno EN, Zuur AF. A Beginners’s Guide to Data Exploration and Visualisation with R. Newburgh, United Kingdom: Highland Statistics Ltd.; 2015.

[Editors’ note: what follows is the authors’ response to the second round of review.]

Essential revisions:1. The relevant data to standardize species richness may not be available from the primary literature. However, it should be possible to employ relevant standardization methods within the 5{degree sign} latitudinal bands in which the data have been aggregated. An analysis based on standardized data, at least for the more data-rich latitudinal bands, must be added.

Following a complete re-analysis of the original data, we no longer aggregate a-diversity within 5° latitudinal bands. The use of 5° latitudinal bands provided a too coarse environmental resolution; local conditions vary substantially between sites in different oceans/regions despite being located within the same 5° of latitude. We therefore extracted environmental data from 1° x 1° latitudinal x longitudinal cells, and allocated the corresponding environmental value to sites located within each cell (please see the much revised method section, and our reply to point 2 below). This approach provides a much more detailed dataset.

Consequently, we do not standardize data across 5° latitudinal bands, and in the discussion, we highlight that patterns of local a-diversity cannot be compared with patterns in regional diversity where other processes (e.g. upwelling, ocean currents) may be important driving factors. We do agree that more studies on global patterns in regional intertidal diversity is needed, but this is beyond the scope of our contribution.

2. Employ models that allow assessing unimodality, which is stated but untested. At the bare minimum, a quadratic relationship with latitude should be included in the GLMM. As implemented here, the GLMM employed to relate diversity to latitude can only detect linear trends, but not unimodal patterns and the mid-latitude peak suggested by LOESS for the northern hemisphere. To provide a formal test for unimodality, models with or without a quadratic term could be contrasted using standard model comparison procedures. Alternatively, GAM could be used to evaluate nonlinear effects.

We thank the reviewers for the good and constructive advice on the statistical protocol.

We acknowledge the issue of a non-linear pattern, and we have re-analyzed the dataset. The residuals distribution in the northern hemisphere dataset did indeed indicate a non-linear pattern, and we re-analysed these data using GAMM. The southern hemisphere dataset is suitable for linear models (e.g. no residuals patterns or other indications), and these data have been analyzed using a GLMM. The whole data exploration process was redone.

Specifically, we started with a full dataset that included latitude, and five latitudinal-scale oceanographic drivers (chlorophyll a, nitrate, phosphate, sea surface temperature and silicate). However, when looking at relationships between co-variates using Pearson’s correlation coefficients, high collinearity was observed between nitrate and phosphate (Pearson r = 0.94), silicate and temperature (Pearson r = 0.73), and oxygen and temperature (Pearson r = 0.99). Based the variance inflation factor (VIF; threshold <10 (Montgomery & Peck, 1992)) we excluded nitrate, silicate and oxygen from the models. In the reduced models was there no collinearity. Thus the final models included chlorophyll a, phosphate and sea surface temperature.

The GAMM and GLMM was fitted with a negative binomial distribution to ensure acceptable residual patterns and to avoid over-dispersion. Both models included the factor ‘country’ as a random effect. For the GAMM, a cubic spline regression smoother was fitted for latitude, and the optimal amount of smoothing was selected using cross-validation (Zuur et al., 2014). The final model was validated by plotting residuals versus fitted values, versus each covariate in the model, and versus each covariate not in the model, following the protocal decribed by (Zuur et al., 2014; Zuur & Ieno, 2016). Bubble plots and variogram were used to inspect model residuals for spatial correlations. They showed no indication of spatial autocorrelation among sites. Final model validation showed no residuals patterns or violations, and we used both models to analyse latitudinal patterns.

Because of these extensive changes, the entire ‘statistical analysis’ section has been rewritten. Please read the revised method section.

3. Clarify whether p-values are relevant or not. As is, it is confusing. For example, the legend of Table 1 mentions p-values, but these are not reported. Materials and methods indicate that 95% confidence intervals are used to take decisions on null hypotheses, suggesting that p-values are not used in the analysis (lines 436-439). Nevertheless, p-values are reported in Table 2.

In the revised manuscript we present p-values for all analyses

4. Provide a rationale for distinguishing between canopy and other algal forms (the distinction is compelling, but it is not explained).

We have explained why we only focus on canopy forming. Please see a detailed response to this issue under point 8 below.

5. We like the conclusion on the importance of local-scale processes. This should be placed in the context of previous studies that have quantified patterns and processes at multiple scales reaching the same conclusion.

We have revised large sections of the discussion to better place our work in context of previous studies on small-scale processes. Please see the revised discussion.

6. We could not access the data repository indicated in ref. 91, so we could not assess whether the analysis may have missed potentially relevant papers.

We will make the data repository public upon acceptance of the manuscript. We have submitted the list of data manuscripts/databases for the reviewers to see if we missed any papers they find relevant. All data will be freely available upon acceptance.

7. Provide the number of studies available for each band in an Appendix.

This has been done, and there is a reference in the Results section:

“Only 11 sites were located at latitudes above 70°, south and north combined (Appendix 1)” (Lines 102-103)

8. The analysis on macroalgae (e.g., Figure 5) distinguished between canopy and no-canopy algae. This is probably correct, but the rationale for this distinction has not been provided. I think some context is needed, especially to clarify the role of algal canopies in maintaining diversified understory assemblages.

We have clarified the important role of intertidal algal canopies in the introduction:

“Macroalgal canopies shelter understory species from environmental stress (Krause-Jensen et al., 2016; Sejr et al., 2021), creating protective microhabitats that increase environmental heterogeneity and biodiversity, hereby maintaining a diversified understory assemblages (Bulleri et al., 2002; Watt & Scrosati, 2013; Piazzi et al., 2018).” (lines 54-58)

In the result section:

“We focused exclusively on canopy-forming algae because canopies provide living space and protection from predation and extreme temperatures, thereby increasing a-diversity and coverage of understory organisms” (lines 118-120)

In the method section:

“We focused on ‘macroalgal cover’ of canopy-forming algae (i.e. non-canopy algae were excluded from this dataset) because canopies creates understory living space and a predation refugium that increase a-diversity (Benedetti-Cecchi et al., 2001; Bulleri et al., 2002; Watt & Scrosati, 2013).” (lines 468-470)

9. The overall conclusion that more studies are needed to assess the magnitude and influence of physical and biotic drivers across multiple scales is important and appropriate. However, many studies have examined processes across scales in rocky intertidal systems (including canopy-removal and limpet-exclusion experiments) and many descriptive studies have quantified variation across multiple spatial scales, emphasizing the importance of small-scale variability in pattern of distribution, abundance and diversity of species on rocky shores. A more thorough discussion of this literature (e.g., Underwood & Chapman, Benedetti-Cecchi, Denny, Coleman, Martins, Fraschetti, etc) would be welcome.

We have added a more in-depth discussion of the importance of small-scale variability on rocky shores. Please see the revised discussion.

10. In the abstract please say something about more sampling being needed in the tropics. Perhaps line 34.

We have added the following sentence addressing sampling in the tropics:

“Polar and tropical intertidal data were sparse, and more sampling in tropical regions is required to improve knowledge of marine biodiversity. (lines 36-37).

11. Line 85: What do you mean by "inadequate estimates of intertidal areas"? Inadequate in what way? And in what sense are you talking about area?

What we meant to say was, that the geographic area (in km^2^) of intertidal zones hasn’t been estimated on a global scale. In 2019, Murray et al. mapped for the first time the extent of tidal flats, but the extent of other intertidal habitats remain unknow.

We have clarified this in the text:

“Global assessments of intertidal biodiversity have been hindered by data scarcity from polar intertidal shores, and a lack of estimates of intertidal geographic areas. In fact only the extent of intertidal mud flats have been quantified on a large scale (Murray et al., 2019).” (lines 84-86)

12. Line 96: Replace "controlled" by "predicted"

Done.

13. Figure 2: where are the R2 values on this plot? How do you get an R2 from a LOESS fit…?

This was an unfortunate typo. There are no R^2^ values presented on Figure 2. We have updated the figure legend:

“Figure 2: Latitudinal patterns in rocky intertidal a-diversity plotted against latitude. Data are split into southern and northern hemisphere. A linear regression line (southern hemisphere) and a best-fit locally weighted scatterplot smoother (northern hemisphere) was added with 95% confidence intervals to aid visual interpretation” (lines 958-961)

14. Salinity is a local and a regional variable in your analyses, please briefly explain why (Figure 3 and 4).

We have given this much thought, and we have decided to exclude salinity as a latitudinal factor. We originally included this parameter as salinity usually reaches maximum values around latitudes 20°N and 20°S, and decreases toward high latitudes. However, the latitudinal data is obtained from satellite data, and coastal salinity is too strongly affected by coastal processes (such as precipitation and melt water run off; Valiela et al., 2012; Meire et al., 2017; Sejr et al., 2017; Mouginot et al., 2019; Covernton & Harley, 2020; Duarte et al., 2020; Navarro et al., 2020). Thus, satellite derived salinity data may provide limited ecological relevant data, and we wanted to avoid making non-logical conclusions on the impacts of salinity, which is a key abiotic driver for ecological patterns (as we show on the local-scale analysis)

15. Figure 5: hard to see the box plots, consider making them a different colour or shade.

We agree. The box plots have been highlighted and superimposed on the data points (see revised Figure 3).

References:

Benedetti-Cecchi, L., Pannacciulli, F.G., Bulleri, F., Moschella, P.S., Airoldi, L., Relini, G. & Cinelli, F. (2001) Predicting the consequences of anthropogenic disturbance: large-scale effects of loss of canopy algae on rocky shores. *Marine Ecology Progress Series*, 214, 137– 150.

Bulleri, F., Benedetti-Cecchi, L., Acunto, S., Cinelli, F. & Hawkins, S.J. (2002) The influence of canopy algae on vertical patterns of distribution of low-shore assemblages on rocky coasts in the northwest Mediterranean. *Journal of Experimental Marine Biology and Ecology*, 267, 89– 106.

Covernton, G. & Harley, C. (2020) Multi-scale variation in salinity: a driver of population size and structure in the muricid gastropod Nucella lamellosa. *Marine Ecology Progress Series*, 643, 1– 19.

Duarte, C.M., Rodriguez-Navarro, A.B., Delgado-Huertas, A. & Krause-Jensen, D. (2020) Dense Mytilus Beds Along Freshwater-Influenced Greenland Shores: Resistance to Corrosive Waters Under High Food Supply. *Estuaries and Coasts*, 43, 387–395.

Krause-Jensen, D., Marbà, N., Sanz-Martin, M., Hendriks, I., Thyrring, J., Carstensen, J., Sejr, M.

& Duarte, C. (2016) Long photoperiods sustain high pH in arctic kelp forests. *Science Advances*, 2, e1501938.

Meire, L., Mortensen, J., Meire, P., Juul-Pedersen, T., Sejr, M.K., Rysgaard, S., Nygaard, R., Huybrechts, P. & Meysman, F.J.R. (2017) Marine-terminating glaciers sustain high productivity in Greenland fjords. *Global Change Biology*, 23, 5344–5357.

Montgomery, D.C. & Peck, E.A. (1992) *Introduction to Linear Regression Analysis*, WileyInterscience.

Mouginot, J., Rignot, E., Bjørk, A.A., van den Broeke, M., Millan, R., Morlighem, M., Noël, B., Scheuchl, B. & Wood, M. (2019) Forty-six years of Greenland Ice Sheet mass balance from 1972 to 2018. *Proceedings of the National Academy of Sciences of the United States of America*, 116, 9239–9244.

Murray, N.J., Phinn, S.R., DeWitt, M., Ferrari, R., Johnston, R., Lyons, M.B., Clinton, N., Thau, D. & Fuller, R.A. (2019) The global distribution and trajectory of tidal flats. *Nature*, 565, 222– 225.

Navarro, J.M., Détrée, C., Morley, S.A., Cárdenas, L., Ortiz, A., Vargas-Chacoff, L., Paschke, K., Gallardo, P., Guillemin, M.L. & Gonzalez-Wevar, C. (2020) Evaluating the effects of ocean warming and freshening on the physiological energetics and transcriptomic response of the Antarctic limpet Nacella concinna. *Science of the Total Environment*, 748, 142448.

Piazzi, L., Bonaviri, C., Castelli, A., Ceccherelli, G., Costa, G., Curini-Galletti, M., Langeneck, J., Manconi, R., Montefalcone, M., Pipitone, C., Rosso, A. & Pinna, S. (2018) Biodiversity in canopy-forming algae: Structure and spatial variability of the Mediterranean Cystoseira assemblages. *Estuarine, Coastal and Shelf Science*, 207, 132–141.

Sejr, M.K., Mouritsen, K.N., Krause-Jensen, D., Olesen, B., Blicher, M.E. & Thyrring, J. (2021) Small scale factors modify impacts of temperature, ice scour and waves and drive rocky intertidal community structure in a Greenland fjord. *Frontiers in Marine Science*, 7, 607135. Sejr, M.K., Stedmon, C.A., Bendtsen, J., Abermann, J., Juul-Pedersen, T., Mortensen, J. & Rysgaard, S. (2017) Evidence of local and regional freshening of Northeast Greenland coastal waters. *Scientific Reports*, 7, 13183.

Valiela, I., Camilli, L., Stone, T., Giblin, A., Crusius, J., Fox, S., Barth-Jensen, C., Monteiro, R.O., Tucker, J., Martinetto, P. & Harris, C. (2012) Increased rainfall remarkably freshens estuarine and coastal waters on the Pacific coast of Panama: Magnitude and likely effects on upwelling and nutrient supply. *Global and Planetary Change*, 92–93, 130–137.

Watt, C.A. & Scrosati, R.A. (2013) Regional consistency of intertidal elevation as a mediator of seaweed canopy effects on benthic species richness, diversity, and composition. *Marine Ecology Progress Series*, 491, 91–99.

Zuur, A.F. & Ieno, E.N. (2016) A protocol for conducting and presenting results of regression-type analyses. *Methods in Ecology and Evolution*, 7, 636–645.

Zuur, A.F., Saveliev, A.A. & Ieno, E.N. (2014) *A beginner’s Guide to Generalised Additive Mixed models with R*, Highland Statistics Ltd., Newburgh, United Kingdom.